# Unpicking Data at the Seams: VAEs, Disentanglement and Independent Components

## Abstract

Disentanglement, or identifying salient statistically independent factors of the data, is of interest in many areas of machine learning and statistics, with relevance to synthetic data generation with controlled properties, robust classification of features, parsimonious encoding, and a greater understanding of the generative process underlying the data. Disentanglement arises in several generative paradigms, including Variational Autoencoders (VAEs), Generative Adversarial Networks and diffusion models. Particular progress has recently been made in understanding disentanglement in VAEs, where the choice of diagonal posterior covariance matrices is proposed to promote mutual orthogonality between columns of the decoder's Jacobian. We continue this thread to show how this *linear* independence translates to *statistical* independence, completing the chain in understanding how the VAE's objective identifies independent components of, or disentangles, the data.

## 1 Introduction

Variational Autoencoders (**VAE**s, Kingma (2013); Rezende et al. (2014)) and a range of variants, e.g. $\beta$-VAE (e.g. Higgins et al., 2017) and Factor-VAE (Kim & Mnih, 2018), have been shown empirically to *disentangle* latent factors of variation in the data. For example, a trained VAE may generate face images that vary in distinct semantically meaningful ways, such as hair colour or facial expression, as individual latent variables are adjusted. This is both of practical use, e.g. for controlled generation of synthetic data with chosen properties, and intriguing as it is not knowingly designed into the training algorithm. A related phenomenon is observed in samples from a Generative Adversarial Network (GAN), which, in common with a VAE, applies a deterministic neural network function to samples of independently distributed latent variables, producing a *push-forward* distribution.

Understanding why disentanglement arises, seemingly "for free", is of interest since identifying and separating generative factors underlying the data goes to the heart of many aspects of machine learning, from classification to generation, interpretability to identifiability, right down to a fundamental understanding of the data itself. With a better appreciation of why disentanglement happens, we might be able to induce it more reliably, particularly in domains where we cannot easily perceive when features are disentangled, as we can for images and text.

Research into the cause of disentanglement has gradually led to a refined understanding of what is meant by "disentanglement", which typically refers to the separation of semantically meaningful generative factors (Bengio et al., 2013). Recent progress has been made towards understanding why disentanglement occurs in VAEs, tracing the root cause to the common use of *diagonal posterior covariance matrices*, a seemingly innocuous design choice made for computational efficiency (Rolinek et al., 2019; Kumar & Poole, 2020). Diagonal covariances are shown to promote orthogonality between columns of the Jacobian of the decoder, a property linked to disentangled features (Ramesh et al., 2018) and independent causal mechanisms (Gresele et al., 2021). We extend this line of work by providing a firmer basis for the covariance-orthogonality relationship together with theoretical analysis to show how orthogonality in the Jacobian translates to disentanglement in the push-forward distribution of a VAE, connecting *linear* independence of partial derivatives to *statistical* independence of components, or generative factors, of the data.

Interest in understanding how disentanglement arises in VAEs has increased as their generative quality has improved (e.g. Hazami et al., 2022) and they often a key component in state of the art diffusion

models, where disentanglement is of great interest (Pandey et al., 2022; Zhang et al., 2022; Yang et al., 2023).

In this work, we analyse and extend recent advances in understanding disentanglement in VAEs by

- proving that orthogonality, or *linear independence*, between columns of the decoder Jacobian corresponds to identifying *statistically independent* components of the generative distribution with distinct latent variables (§**??**);
- providing conditions under which a VAE fully identifies the data distribution (§**??**); and
- presenting a novel interpretation of $\beta$ in a $\beta$-VAE, as scaling the variance of the likelihood distribution, explaining why $\beta$ affects both disentanglement and "posterior collapse" (§5).

## 2 BACKGROUND

**Disentanglement**: Disentanglement is not consistently defined in the literature, but typically refers to identifying salient, semantically meaningful features of the data with distinct latent variables, such that by varying a single variable, data can be generated that differ in a single aspect (Bengio et al., 2013; Higgins et al., 2017; Ramesh et al., 2018; Rolinek et al., 2019). Disentanglement has also been decomposed into necessary and sufficient-type concepts of *consistency* and *restrictiveness* (Shu et al., 2019). We show that disentanglement in a VAE relates to identifying *statistically independent components* of the data, comparable to independent component analysis (**ICA**).

**Variational Autoencoder (VAE)**: A VAE is a latent generative model for data $x \in \mathcal{X} \doteq \mathbb{R}^m$, that models the data distribution by $p_\theta(x) = \int_z p_\theta(x|z)p(z)$ with parameters $\theta$ and latent variables $z \in \mathcal{Z} \doteq \mathbb{R}^d$. A VAE is trained by maximising a lower bound to the log likelihood (the **ELBO**),

$$\int_x p(x) \log p_\theta(x) \quad \geq \quad \int_x p(x) \int_z q_\phi(z|x) \big\{ \log p_\theta(x|z) - \beta \log \tfrac{q_\phi(z|x)}{p(z)} \big\} \quad \doteq \ell(\theta, \phi) \,, \quad (1)$$

where $q_\phi(z|x)$ learns to approximate the model posterior $p_\theta(z|x) \doteq \frac{p_\theta(x|z)p(z)}{p_\theta(x)}$; and $\beta = 1$. A VAE parameterises distributions by neural networks: $q_\phi(z|x) = \mathcal{N}(z; e(x), \Sigma_x)$ has mean $e(x)$ and diagonal covariance $\Sigma_x$ output by an *encoder* network; and $p_\theta(x|z)$ is typically of exponential family form (e.g. Bernoulli or Gaussian) with natural parameter $\theta \doteq d(z)$ defined by a *decoder* network.[1] The prior is commonly a standard Gaussian $p(z) = \mathcal{N}(z; \mathbf{0}, \mathbf{I})$.

While samples generated from a VAE ($\beta = 1$) can exhibit disentanglement, setting $\beta > 1$ is found empirically to enhance the effect, typically with a cost to generative quality (Higgins et al., 2017).

**Probabilistic Principal Component Analysis (PPCA)**: PPCA (Tipping & Bishop, 1999) considers a *linear* latent variable model with parameters $\mathbf{D} \in \mathbb{R}^{m \times d}$, $\sigma \in \mathbb{R}$ and noise $\epsilon \in \mathbb{R}^m$.[2]

$$x = \mathbf{D}z + \epsilon \qquad z \sim p(z) = \mathcal{N}(z; \mathbf{0}, \mathbf{I}) \qquad \epsilon \sim p_\sigma(\epsilon) = \mathcal{N}(\epsilon; \mathbf{0}, \sigma^2 \mathbf{I}) \,, \quad (2)$$

All distributions are Gaussian and known analytically, in particular the model posterior is given by

$$p_\theta(z|x) = \mathcal{N}(z; \tfrac{1}{\sigma^2} \mathbf{M} \mathbf{D}^\top x, \, \mathbf{M}) \qquad \text{where} \quad \mathbf{M} = (\mathbf{I} + \tfrac{1}{\sigma^2} \mathbf{D}^\top \mathbf{D})^{-1} \,. \quad (3)$$

The maximum likelihood solution is fully tractable: $\mathbf{D}_{\text{PPCA}} = \mathbf{U}(\mathbf{S} - \sigma^2 \mathbf{I})^{1/2} \mathbf{R}$, where $\mathbf{S} \in \mathbb{R}^{d \times d}$ and $\mathbf{U} \in \mathbb{R}^{m \times d}$ contain the largest eigenvalues and corresponding eigenvectors of the data covariance $\mathbf{X} \mathbf{X}^\top$, and $\mathbf{R} \in \mathbb{R}^{d \times d}$ is orthonormal ($\mathbf{R}^\top \mathbf{R} = \mathbf{I}$). As $\sigma^2 \to 0$, $\mathbf{D}_{\text{ML}}$ approaches the singular value decomposition (**SVD**) of the data $\mathbf{X} = \mathbf{U} \mathbf{S} \mathbf{V}^\top$ up to ambiguity in $\mathbf{V}$, as in classical PCA. Due to the ambiguity in $\mathbf{R}/\mathbf{V}$, the model is considered *unidentified*. While the exact solution is known, it can also be numerically approximated by optimising the ELBO, e.g. (Eq. 1) by *expectation maximisation* (**PPCA**$^{EM}$), where (**E**) sets $q_\phi(z|x)$ to its exact optimum $p_\theta(z|x)$ in Eq. 3; and (**M**) optimises w.r.t. $\theta$.

**Linear VAE (LVAE)**: A VAE with Gaussian likelihood $p_\theta(x|z) \doteq \mathcal{N}(x; d(z), \sigma^2 \mathbf{I})$ and linear decoder $d(x) = \mathbf{D}x$ (termed a *linear VAE* assumes the same underlying model as PPCA (2). Indeed, training an LVAE differs to PPCA$^{EM}$ only in approximating the posterior by $q_\phi(z|x) = \mathcal{N}(z; \mathbf{E}x, \Sigma)$

---

[1] For Gaussian distributions, a fixed variance parameter $\sigma^2$ is also specified.

[2] Throughout, we assume data is centred which equates to including a mean parameter (Tipping & Bishop, 1999).

rather than computing its analytic optimum. While the latter may seem preferable, Lucas et al. (2019) showed that an LVAE with diagonal $\Sigma$ *breaks the symmetry* of PPCA. This follows from $\Sigma$ being both diagonal and optimal per Eq. 3,

$$\Sigma \;=\; \boldsymbol{M}_{\text{PPCA}} \;\doteq\; (\boldsymbol{I} + \tfrac{1}{\sigma^2}\boldsymbol{D}_{\text{PPCA}}^{\top}\boldsymbol{D}_{\text{PPCA}})^{-1} \;=\; \sigma^2 \boldsymbol{R}^{\top}\boldsymbol{S}^{-1}\boldsymbol{R} \qquad \forall x, \tag{4}$$

(by definition of $\boldsymbol{D}_{\text{PPCA}}$). This requires $\boldsymbol{R} = \boldsymbol{I}$ and restricts the solution of an LVAE to $\boldsymbol{D}_{\text{LVAE}} = \boldsymbol{U}(\boldsymbol{S} - \sigma^2\boldsymbol{I})^{1/2}$ (*cf* $\boldsymbol{D}_{\text{PPCA}}$), up to trivial transformations (axis permutation and sign).

**Orthogonality in a VAE Decoder's Jacobian**: Beyond symmetry breaking in *linear* VAEs, diagonal posterior covariances are shown to promote disentanglement in *non-linear* VAEs by inducing columns of the decoder's Jacobian to be mutually orthogonal (Rolinek et al., 2019; Kumar & Poole, 2020). The generalised argument of Kumar & Poole (2020) reparameterises around the encoder mean, $z = e(x) + \epsilon$, $\epsilon \sim \mathcal{N}(\boldsymbol{0}, \Sigma_x)$, and Taylor expands to approximate a *deterministic* ELBO (**det-ELBO**):

$$\ell(x) = \mathbb{E}_{\epsilon|x}\Big[ \log p_\theta(x|z = e(x) + \epsilon) \; - \; \beta \underbrace{\log \tfrac{p(\epsilon)}{p(z = e(x) + \epsilon)}}_{\text{KL}} \Big] \tag{Reparameterise}$$

$$= \mathbb{E}_{\epsilon|x}\Big[ \log p_\theta(x|z = e(x)) \; + \; \epsilon^{\top}\boldsymbol{j}_{e(x)}(x) \; + \; \tfrac{1}{2}\epsilon^{\top}\boldsymbol{H}_{e(x)}(x)\epsilon + O(\epsilon^3) \; - \; \beta\,\text{KL} \Big] \tag{Taylor}$$

$$\approx \underbrace{\log p_\theta(x|z = e(x))}_{\text{AE}} + \tfrac{1}{2}\underbrace{\boldsymbol{H}_{e(x)}(x) \odot \Sigma_x}_{\text{gradient regularisation}} - \tfrac{\beta}{2}\big( \underbrace{\|e(x)\|^2 + \text{tr}(\Sigma_x)}_{\textbf{prior}} \; - \; \underbrace{\log|\Sigma_x| - d}_{\textbf{entropy}} \big). \tag{5}$$

Here, $\boldsymbol{j}_{z^*}(x) \doteq (\tfrac{\partial \log p_\theta(x|z)}{\partial z_i})_i$ and $\boldsymbol{H}_{z^*}(x) \doteq (\tfrac{\partial^2 \log p_\theta(x|z)}{\partial z_i \partial z_j})_{i,j}$ are the Jacobian and Hessian of $\log p_\theta$ evaluated at $z^* \in \mathcal{Z}$; and $\odot$ is the Frobenius (dot) product. Notably, $\mathbb{E}_{\epsilon|x}[O(\epsilon^3)]$ terms are dropped. Differentiating Eq. 5 w.r.t. $\Sigma_x$ suggests a connection between the Hessian and encoder variance:

$$\nabla_{\Sigma_x}\ell(x) \approx \tfrac{1}{2}\big( \boldsymbol{H}_{e(x)}(x) - \beta(\boldsymbol{I} - \Sigma_x^{-1}) \big) \quad \Rightarrow \quad \boxed{\Sigma_x^{-1} \approx \boldsymbol{I} \;-\; \tfrac{1}{\beta}\boldsymbol{H}_{e(x)}(x)}. \tag{6}$$

As in the linear case, the ELBO with diagonal $\Sigma_x$ is minimised if the likelihood's Hessian is also diagonal. For exponential family $p_\theta(x|z)$ with natural parameter $\theta = d(z)$ defined by the decoder,

$$\boldsymbol{H}_{e(x)}(x) \;=\; -\boldsymbol{D}_{e(x)}^{\top}\boldsymbol{A}_{d\circ e(x)}^2 \boldsymbol{D}_{e(x)} \;+\; (x - \hat{x}_{d\circ e(x)})^{\top}\boldsymbol{\mathsf{D}}_{e(x)}, \tag{7}$$

where $\boldsymbol{D}_z$ / $\boldsymbol{\mathsf{D}}_z$ are the Jacobian / Hessian of the decoder evaluated at $z \in \mathcal{Z}$, $\hat{x}_\theta = \mathbb{E}[x|\theta]$, and $\boldsymbol{A}_\theta^2 = -\tfrac{d^2}{d\theta^2}\log p_\theta(x|z) = \text{Var}[x|\theta]$. $\boldsymbol{A}_\theta$ is diagonal if dimensions $x_i$ of $x$ are conditionally independent given $\theta$, e.g. if $p_\theta(z|x)$ is Gaussian, $\boldsymbol{A}_\theta = \tfrac{1}{\sigma}\boldsymbol{I}$. The key conclusion is that for Gaussian likelihoods and commonly used decoders where the last term in Eq. 7 is small almost everywhere (e.g. ReLU networks), $\Sigma_x^{-1} \approx \boldsymbol{I} + \tfrac{1}{\beta\sigma^2}\boldsymbol{D}_{e(x)}^{\top}\boldsymbol{D}_{e(x)}$, meaning *columns of the decoder Jacobian $\boldsymbol{D}_z$ are orthogonal*.

# 3 FROM DIAGONAL POSTERIOR COVARIANCE TO ORTHOGONALITY

Before building on it, we first make *precise* the relationship between posterior covariance and the log likelihood's Hessian in Eq. 6. Maximising the ELBO with a Gaussian posterior approximation is equivalent to an *averaged Laplace approximation* (Opper & Archambeau, 2009). Hence at optimality,

$$\Sigma_x^{-1} \;\; = -\tfrac{d}{d\Sigma_x}\mathbb{E}_{q(z|x)}[\log p_\theta(x,z)] \;\; = -\mathbb{E}_{q(z|x)}[\tfrac{d^2}{dz^2}\log p_\theta(x,z)] \;\; = \boldsymbol{I} - \tfrac{1}{\beta}\mathbb{E}_{q(z|x)}[\boldsymbol{H}_z(x)], \tag{8}$$

which follows from: (i) differentiating the ELBO w.r.t. $\Sigma_x$; (ii) the link to Laplace approximation; and (iii) the Gaussian prior. (This also follows from the deterministic ELBO (Eq. 5) by differentiating the *full* Taylor series w.r.t. $\Sigma_x$.) Thus, Eq. 6 holds *in expectation*. Accordingly, for Gaussian likelihoods and controlled higher decoder derivatives (e.g. ReLU networks or similar), columns of the decoder Jacobian are *orthogonal in expectation* over each posterior $q_\phi(z|x)$. Although weaker than Eq. 6 suggests, this relationship between column-orthogonality of the decoder Jacobian and disentanglement is observed empirically (Rolinek et al., 2019; Kumar & Poole, 2020). We conjecture that orthogonality holds more consistently the more posteriors overlap and regions of overlap are subject to multiple simultaneous orthogonality constraints (see §5). This less rigid relationship may also partly justify why disentanglement is observed variably in practice (e.g. Locatello et al., 2019).

Comparing optimal covariances for PPCA (Eq. 3) and Gaussian VAE (dropping $\beta$ for clarity),

$$\hat{\Sigma}_{\text{PPCA}}^{-1} = \boldsymbol{I} + \tfrac{1}{\sigma^2}\boldsymbol{D}^\top\boldsymbol{D} \qquad \hat{\Sigma}_{x,\text{VAE}}^{-1} = \mathbb{E}_{q(z|x)}[\boldsymbol{I} + \tfrac{1}{\sigma^2}\boldsymbol{D}_{e(x)}^\top\boldsymbol{D}_{e(x)}]\,, \tag{9}$$

reveals how the optimal posterior covariance for a non-linear VAE generalises the well-known result for the linear case. This is insightful, since it is this relationship that enables an LVAE to break the rotational symmetry of PPCA (Eq. 4) and, more pertinently, results in standard basis vectors $\boldsymbol{z}_i \in \mathcal{Z}$ corresponding to, or *identifying*, independent principal axes of variance of the data: $\boldsymbol{D}_{\text{LVAE}}\boldsymbol{z}_i = \boldsymbol{U}(\boldsymbol{S} - \sigma^2\boldsymbol{I})^{1/2}\boldsymbol{z}_i \propto \boldsymbol{u}_i$. In short, the relationship that *disentangles* independent factors of variation in the linear case, is mirrored in the non-linear case.

Note (for future reference, §5) that higher $\mathrm{Var}[x|z] = \sigma^2$ corresponds to higher $\mathrm{Var}[z|x] = \Sigma$, and vice versa, i.e. uncertainty in one domain goes hand in hand with uncertainty in the other.

**Linear det-ELBO**: Lastly, we comment briefly on Eq. 5 for an LVAE (dropping $\beta$ for clarity),

$$2\ell^{\text{LVAE}} = \mathbb{E}_x\big[-\tfrac{1}{\sigma^2}\|x - \boldsymbol{D}\boldsymbol{E}x\|^2 \; - \; (\boldsymbol{I} + \tfrac{1}{\sigma^2}\boldsymbol{D}^\top\boldsymbol{D}) \odot \Sigma \; + \; \log|\Sigma| \; - \; \|\boldsymbol{E}x\|^2 \; + \; d\big]$$
$$= \mathbb{E}_x\big[-\tfrac{1}{\sigma^2}\|x - \boldsymbol{D}\boldsymbol{E}x\|^2 \; - \; \|\boldsymbol{E}x\|^2\big] \; - \; \log|\boldsymbol{I} + \tfrac{1}{\sigma^2}\boldsymbol{D}^\top\boldsymbol{D}|\,, \tag{10}$$

where we plug-in the optimal posterior covariance (Eq. 9). This gives a deterministic objective for PPCA, that can also be seen as a regularised objective for deterministic PCA *that removes ambiguity from $\boldsymbol{R}$* (Eq. 4). This is of interest as variations of PCA are widely studied in terms of their optima (Kunin et al., 2019; Bao et al., 2020) and learning dynamics (Saxe et al., 2014; Bao et al., 2020).

## 4 FROM ORTHOGONALITY TO DISENTANGLEMENT

Having clarified the connection between diagonal posterior covariance and column-orthogonality of the Jacobian, we now develop our main result to show why such orthogonality causes disentanglement. This relates two different notions of "independence": orthogonality pertains to *linear independence*, a geometric property, while disentanglement relates to *statistical independence* of factors/components of a distribution. These concepts do not always go hand-in-hand and it is not immediately clear why a column-orthogonal Jacobian implies that different latent dimensions correspond to statistically independent, often semantically meaningful, factors of variation in the data.

The generative model of a VAE can be decomposed into stochastic and deterministic steps: sample the prior $z \sim p(z)$; apply a deterministic function $\hat{x} = \mu \circ d(z)$ (composing the decoder $d: z \mapsto \theta$ and mean function $\mu: \theta \mapsto \hat{x} \doteq \mathbb{E}[x|z]$); and add element-wise noise, $x \sim p(x|\hat{x})$. For continuous data, e.g. images, element-wise noise often serves only as "blur" and is omitted when generating synthetic data. Thus, samples come from the "push-forward" distribution of mean parameters $p(\hat{x})$ over a $d$-dimensional manifold defined by the decoder. Note that if data are truly generated under the VAE model for some ground truth decoder parameters $\theta^*$ and small $\sigma$,[3] then as $p_\theta(x) \overset{\mathcal{D}}{\to} p(x)$, the model push-forward distribution tends to the ground truth push-forward distribution, $p_\theta(\hat{x}) \overset{\mathcal{D}}{\to} p(\hat{x})$ (proof: by contradiction). We now consider such push-forward distributions ($\mathcal{X} = \mathbb{R}^m$, $\mathcal{Z} = \mathbb{R}^d$, $d \leq m$).

**Definition 1** (push-forward distribution). *For a function $f : \mathcal{Z} \to \mathcal{X}$ and prior distribution over $\mathcal{Z}$, $p_z(z)$, the push-forward distribution $p_{f,p_z}^\#(x)$ is defined implicitly over $\{x = f(z) \mid z \sim p(z)\} \subseteq \mathcal{X}$.*

Unless stated otherwise, we assume:

**A#1.** *Latent variables are sampled from independent standard normals, $p_z(z) = \prod_{i=1}^d \mathcal{N}(z_i; 0, 1)$.*[4]

**A#2.** *$f : \mathcal{Z} \to \mathcal{X}$ is injective, continuous and differentiable almost everywhere.*[5]

Note that under A#2, $f$:

(i) defines a $d$-dimensional *manifold* $\mathcal{M}_f = \{f(z) \mid z \in \mathcal{Z}\} \subseteq \mathcal{X}$ embedded in $\mathbb{R}^m$ supporting $p_f^\#$;

(ii) is bijective between $\mathcal{Z}$ and $\mathcal{M}_f$; and

(iii) has full-rank Jacobian $\boldsymbol{J}$ evaluated at $z^*$, $\forall z^* \in \mathcal{Z}$ (by injectivity).

---

[3]Relative to variance of the mean parameters over the manifold, in effect the signal-to-noise-ratio.

[4]Since we assume A#1 throughout, $p_Z(z)$ is generally dropped from the subscript of $p_f^\#$ to lighten notation.

[5]We refer to this as "quasi-differentiable" and note that this includes piece-wise linear functions.

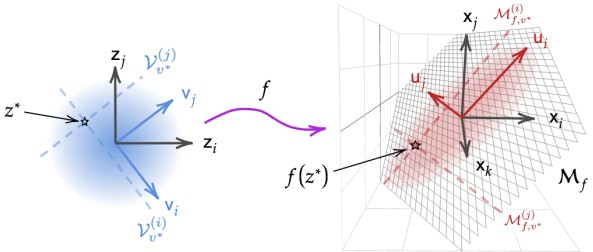

Figure 1: Translating linear independence to statistical independence (**linear $f$**, with Jacobian $J = USV^\top$ and manifold $\mathcal{M}_f \subseteq \mathcal{X}$): A point in $\mathcal{Z}$ is denoted $z$ when in the standard basis $\{z_i, z_j\}$ and $v$ in the $V$-basis (columns of $V$: $v_i, v_j$, solid blue). $\mathcal{V}_{v^*}^{(i)} \subseteq \mathcal{Z}$ are lines passing through $v^*$ as each co-ordinate $v_i$ varies (dashed blue). Each $\mathcal{V}_{v^*}^{(i)}$ maps to a sub-manifold $\mathcal{M}_{f,v^*}^{(i)} \subseteq \mathcal{M}_f$ (dashed red) passing through $f(z^*)$ parallel to the $U$-basis (columns of $U$: $u_i, u_j$, solid red). $\mathcal{V}_{v^*}^{(i)}$ are statistically independent: $p(v^*) = \prod_i p(v_i^*)$. $f$ induces push-forward distributions $p_{f,v^*}^{\#(i)}$ over $\mathcal{M}_{f,v^*}^{(i)}$ and for $x = f_V(v^*) \equiv f(z^*)$, the density over the manifold $p(x) = \prod_i p_{f,v^*}^{\#(i)}(x)$ factorises as a product of independent components.

Letting $J = USV^\top$ denote the SVD of $J$ ( $U^\top U = I$, $V^\top V = VV^\top = I$), we note that columns $v_i \in \mathcal{Z}$ of $V$ define a (local) orthonormal basis for $\mathcal{Z}$, while columns $u_i \in \mathcal{X}$ of $U$ define a basis for the tangent space to $\mathcal{M}_f$ at $f(z^*)$. Since $Jv_i = s_i u_i$ (where $s_i \doteq S_{i,i}$, the $i^{\text{th}}$ singular value), at each $z^* \in \mathcal{Z}$ the Jacobian identifies (local) basis vectors in $\mathcal{Z}$ with (local) basis vectors in $\mathcal{X}$.

## 4.1 LINEAR $f$

For intuition, we first consider the linear case $f(z) = Dz$ (satisfying A#2). Here, $\mathcal{M}_f$ is a $d$-dimensional linear manifold (hyperplane through the origin) describing the mean parameters in PPCA/LVAE. The Jacobian $J = D$ (and SVD components $U, S, V$) is constant $\forall z^* \in \mathcal{Z}$, and the density of the push-forward distribution

$$p_f^\#(x = f(z)) = |D|^{-1} p(z) = \prod_i |s_i|^{-1} p(z_i) \tag{11}$$

factorises. We now interpret factors $|s_i|^{-1} p(z_i)$. We express $z$ in the $V$-basis as $v = V^\top z$ under a transformation with Jacobian $\frac{\partial v}{\partial z} = V^\top$. Thus $p(v) = |V| p(z) = p(z)$ (as expected given only the basis/perspective has changed) and, by rotational symmetry of the Gaussian prior, $p(v) = \prod_i p(v_i)$ and $p(v_i) = \mathcal{N}(0,1)$. We can consider $x$ as a function of $z$ expressed in the $V$-basis (i.e. of $v$), $x = f(z) = USv \doteq f_V(v)$, for which the partial derivatives (columns of $f_V$'s Jacobian) are now orthogonal: $\frac{\partial x}{\partial v_i}^\top \frac{\partial x}{\partial v_j} = s_i s_j u_i^\top u_j = \{s_i^2 \text{ if } i = j, \text{ o/w } 0\}$; and $\|\frac{\partial x}{\partial v_i}\| = |s_i|$. Eq. 11 thus becomes

$$p_f^\#(x = f_V(v)) = \prod |s_i|^{-1} p(v_i) = \prod \|\tfrac{\partial x}{\partial v_i}\|^{-1} p(v_i), \tag{12}$$

where factors now have the form of uni-variate probability distributions under a change of variables. In simple terms, the $V$-basis is a "natural" way to view the data, hence the Jacobian factorises.

With reference to Figure 1, let $v^* = V^\top z^*$ be an evaluation point expressed in the $V$-basis and let $\{\mathcal{V}_{v^*}^{(i)} \subseteq \mathcal{Z}\}_i$ be lines passing through $v^*$ as co-ordinate $i$ in the $V$-basis varies (dashed blue), i.e. $\mathcal{V}_{v^*}^{(i)} = \{(v_1^*, ..., v_i, ..., v_d^*) \mid v_i \in \mathbb{R}\}$. The image of each $\mathcal{V}_{v^*}^{(i)}$ under $f_V$ forms a line (or linear sub-manifold) $\mathcal{M}_{f,v^*}^{(i)} = \{f_V(v) \mid v \in \mathcal{V}_{v^*}^{(i)}\} \subseteq \mathcal{M}_f$ (dashed red). As $\mathcal{V}_{v^*}^{(i)}$ follow right-singular vectors in $\mathcal{Z}$ (the $V$-basis), $\mathcal{M}_{f,v^*}^{(i)}$ follow left-singular vectors in $\mathcal{X}$ (the $U$-basis). Since in the linear case $U, V$ are constant and define global bases (solid red/blue), $\mathcal{M}_{f,v^*}^{(i)}$ are parallel ($\forall v^*$).

Just as we considered $z$ in the $V$-basis, we can consider $x$ in the $U$-basis, $u = U^\top x$, $u_i = u_i^\top x$, to see

$$J = USV^\top = \tfrac{dx}{du}\tfrac{du}{dv}\tfrac{dv}{dz} = U\tfrac{du}{dv}V^\top \quad \Rightarrow \tfrac{du}{dv} = S \quad \Rightarrow p(u) = \prod_i |s_i|^{-1} p(v_i) = \prod_i p(u_i) \tag{13}$$

showing that each component $u_i$ depends only on independent variable $v_i$, so $\{u_i\}_i$ are independent. Since only the basis changes, $p_f^\#(x) = p(u)$ and by comparing Eqs. 12 and 13, we see that the push-forward distribution over the manifold factorises into independent components in the $U$-basis. As the final step, note that the push-forward distributions defined by $f$ restricted to each line $v \in \mathcal{V}_{v^*}^{(i)}$ are supported on sub-manifolds $\mathcal{M}_{f,z^*}^{(i)}$ with density $p_{f,v^*}^{\#(i)}(x) \doteq \|\frac{\partial x}{\partial v_i}\|^{-1} p(v_i) = p(u_i)$. Thus, independent components of $v$ (and so of $z$) are "pushed-forward" to independent components of $u$ (and so of $x$). In summary, for $x = f_v(v) \in \mathcal{M}_f$, the probability density

$$p_f^\#(x) = \prod_i p_{f,v}^{\#(i)}(x) = \prod_i p(u_i) , \qquad (14)$$

factorises as a product of densities over 1-D sub-manifolds in $\mathcal{X}$, analogously to how $p(z)$ factorises into 1-D Gaussians in $\mathcal{Z}$. We state this result formally (with proof steps summarised in A.1) as:

**Theorem 1.** *Assuming A#1 (independent Gaussian latent variables) and a linear function $f : \mathcal{Z} \to \mathcal{X}$, $f(z) = \boldsymbol{D}z$, the push-forward distribution $p_f^\#$ factorises as a product of statistically independent components in $\mathcal{X}$ (Eq. 14). Statistically independent vectors in $\mathcal{Z}$ parallel to right singular vectors of $\boldsymbol{D}$ map to statistically independent vectors in $\mathcal{X}$ parallel to left singular vectors of $\boldsymbol{D}$.*

**Remark 1.** *The probability density over each $\mathcal{V}_{v^*}^{(i)} \subseteq \mathcal{Z}$ is a standard Gaussian $\mathcal{N}(v; 0, 1)$. The density $p_{f,v^*}^{\#(i)}(x) = s_i^{-1} p(f_v^{-1}(x)_i)$ over each sub-manifold, $\mathcal{M}_{f,z^*}^{(i)} \subseteq \mathcal{X}$, is also Gaussian $\mathcal{N}(x; 0, s_i^2)$.*

**Remark 2.** *The SVD of the Jacobian $\boldsymbol{J} = \boldsymbol{U}\boldsymbol{S}\boldsymbol{V}^\top$ can be interpreted in terms of the chain rule $\boldsymbol{J} = \frac{\partial x}{\partial u} \frac{\partial u}{\partial v} \frac{\partial v}{\partial z}$; and as $\boldsymbol{U}, \boldsymbol{V}^\top$ transforming the basis in each domain (termed the independent bases of $f$), and diagonal $\boldsymbol{S} = \frac{\partial u}{\partial v}$ is the Jacobian of $f$ for elements expressed in the independent bases. A basis vector in one domain uniquely affects one basis vector in the other: $\frac{\partial u_i}{\partial v_j} = \{s_i \text{ if } i = j; 0 \text{ o/w}\}$.*

**Remark 3.** *Since right singular vectors $\boldsymbol{V}$ are a basis, or matter of perspective, they have no effect on the model and can never be recovered. Thus PPCA is identified if $\boldsymbol{U}$ and $\boldsymbol{S}$ are recovered.*

**Corollary 1.1.** *For data generated under the linear PPCA model (Eq. 2), an LVAE identifies statistically independent components of the data. If singular values of ground truth $\boldsymbol{D}$ are distinct, the model is uniquely identified (up to column permutation and sign).*

*Proof (sketch, see A.2).* The PPCA model satisfies the assumptions of Theorem 1. Columns of $\boldsymbol{D}_{\text{LVAE}}$ identify left singular vectors of ground truth $\boldsymbol{D}$, which, by Theorem 1, define statistically independent components of the data. Identifiability follows from uniqueness of $p_{f,v^*}^{\#(i)}(x)$. $\qquad\square$

## 4.2 NON-LINEAR $f$, COLUMN-ORTHOGONAL JACOBIAN

Theorem 1 for a linear function may not seem surprising, but notably its proof does not rely on linearity of $f$. We now follow a similar argument for $f$ that may be non-linear, assuming instead

**A#3.** *($\forall z^* \in \mathcal{Z}$) columns of $\boldsymbol{J}$ are mutually orthogonal, i.e. $\frac{\partial x}{\partial z_i}^\top \frac{\partial x}{\partial z_j} = 0, i \neq j$; equivalently $\boldsymbol{V} = \boldsymbol{I}$.*

**Theorem 2.** *Assuming A#1-3, the push-forward distribution $p_f^\#$ factorises as a product of statistically independent components in $\mathcal{X}$ (Eq. 17). At each point $z^* \in \mathcal{Z}$, statistically independent vectors in $\mathcal{Z}$ parallel to the standard basis map to statistically independent vectors in $\mathcal{X}$ parallel to left singular vectors of the Jacobian $\boldsymbol{J}$ evaluated at $z^*$.*

*Proof.* The push-forward distribution of $f$ satisfies

$$p_f^\#(f(z)) = |\boldsymbol{J}|^{-1} p(z) = \prod_i |s_i|^{-1} p(z_i) = \prod_i \|\frac{\partial x}{\partial z_i}\|^{-1} p(z_i) , \qquad (15)$$

equivalent to Eq. 12 but without the need for a change of basis. As illustrated in Figure 2 (*left*) and analogously to the linear case, let $\mathcal{Z}_{z^*}^{(i)} \subset \mathcal{Z}$ denote orthogonal lines passing through $z^*$ parallel to the standard basis, $\mathcal{Z}_{z^*}^{(i)} = \{(z_1^*, ..., z_i, ..., z_d^*) \,|\, z_i \in \mathbb{R}\}$ (dashed blue). To isolate the action of $f$ over each $\mathcal{Z}_{z^*}^{(i)}$, we define $f_{z^*}^{(i)} : \mathcal{Z}_{z^*}^{(i)} \to \mathcal{M}_f$, $f_{z^*}^{(i)}(z_i) \doteq f(z_1^*, \ldots, z_i, \ldots, z_d^*)$, which each map the line $\mathcal{Z}_{z^*}^{(i)}$ to a 1-D sub-manifold $\mathcal{M}_{f,z^*}^{(i)} \doteq \{f_{z^*}^{(i)}(z_i) \,|\, z_i \in \mathbb{R}^d\} \subset \mathcal{M}_f$ passing through $f(z^*)$

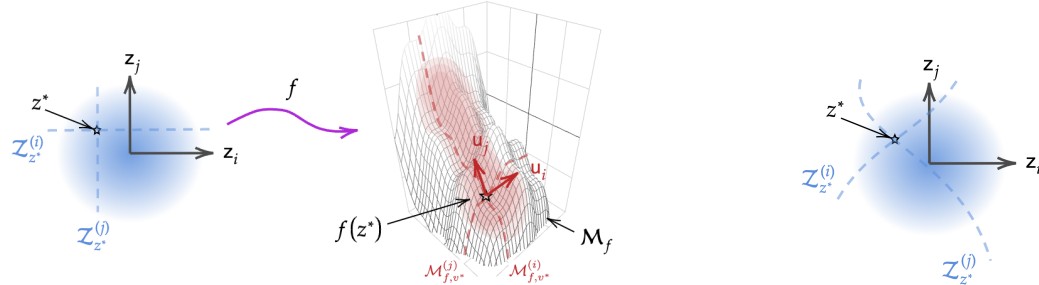

Figure 2: Translating linear independence to statistical independence: (*left*) **non-linear** $f$, orthogonal Jacobian $J = USV^\top$ (evaluated at $z^*$), manifold $\mathcal{M}_f \subseteq \mathcal{X}$: $\mathcal{Z}_{z^*}^{(i)} \subseteq \mathcal{Z}$ are lines passing through $z^*$ by varying co-ordinate $i$ in the standard basis (dashed blue). Each $\mathcal{Z}_{z^*}^{(i)}$ maps to a sub-manifold $\mathcal{M}_{f,z^*}^{(i)} \subseteq \mathcal{M}_f$ (dashed red) passing through $f(z^*)$ parallel to the local $U$-basis (columns of $U$: $u_i, u_j$, red). $\mathcal{Z}_{z^*}^{(i)}$ are statistically independent: $p(z^*) = \prod_i p(z_i^*)$. $f$ induces push-forward distributions $p_{f,z^*}^{\#(i)}$ over $\mathcal{M}_{f,z^*}^{(i)}$ and for $x = f(z^*)$, the density over the manifold $p(x) = \prod_i p_{f,z^*}^{\#(i)}(x)$ factorises as a product of independent components. (*right*) Relaxing the orthogonality requirement: paths defined by following right singular vectors of the Jacobian ($V$) need not be linear and need not correspond to *independent* components of $z$.

(dashed red). Given $f_{z^*}^{(i)}$ is a restriction of $f$, vectors $\frac{d}{dz_i} f_{z^*}^{(i)} = \frac{\partial x}{\partial z_i}$ are tangent to the manifold and sub-manifold $\mathcal{M}_{f,z^*}^{(i)}$ (solid red), and since $\frac{\partial x}{\partial z_i}$ are orthogonal (by assumption), sub-manifolds $\mathcal{M}_{f,z^*}^{(i)}$ are orthogonal at $z^*$. Considering how density is mapped (from dashed blue to dashed red), $f_{z^*}^{(i)}$ together with the marginal $p(z_i)$ over its domain $\mathcal{Z}_{z^*}^{(i)}$ define a push-forward distribution $p_{f,z^*}^{\#(i)}(x) = \|\frac{\partial x}{\partial z_i}\|^{-1} p(z_i)$, over $x = f(z) \in \mathcal{M}_{f,z^*}^{(i)}$. Denoting $x$ in the $U$-basis by $u = U^\top x$,

$$J = US = \frac{dx}{du}\frac{du}{dz} = U\frac{du}{dv} \quad \Rightarrow \frac{du}{dz} = S \quad \Rightarrow p(u) = \prod_i |s_i|^{-1} p(z_i) = \prod_i p(u_i) \quad (16)$$

where $u_i$ are independent. Thus $p(u_i) = \prod_i p_{f,z^*}^{\#(i)}(x)$ and by Eq. 15, for $x \in \mathcal{M}_f$,

$$p_f^{\#}(x) = \prod_i p_{f,z^*}^{\#(i)}(x) = \prod_i p(u_i^\top x) \quad (17)$$

factorises into independent uni-variate distributions, i.e. *statistically independent components*, supported over sub-manifolds $\mathcal{M}_{f,z^*}^{(i)}$ that are mutually orthogonal where they meet. $\qquad \square$

**Remark 4.** *The independent basis of $f$ is the standard basis of $\mathcal{Z}$ and $\|\frac{\partial x}{\partial z_i}\| = |s_i|$.*

**Remark 5.** *The density restricted to $\mathcal{Z}_{v^*}^{(i)}$ is a standard Gaussian $\mathcal{N}(z; 0, 1)$. The density $p_{f,v^*}^{\#(i)}(x) = |s_i|^{-1} p(f^{-1}(x)_i)$ over $\mathcal{M}_{f,z^*}^{(i)}$ is not Gaussian in general since $s_i$ can vary arbitrarily over $x \in \mathcal{M}_{f,z^*}^{(i)}$.*

**Corollary 2.1.** *For data generated from a push-forward distribution $p_f^{\#}$, where $p(z)$ satisfies A#1 and $f$ satisfies A#2 and A#3, a Gaussian VAE identifies statistically independent components of the data with distinct latent dimensions. If ground truth singular values $s_i$ as a function of $z \in \mathcal{Z}_{v^*}^{(i)}$ are unique, the model is fully identified (the analogue of distinct singular values).*

*Proof.* The data distribution and an optimised Gaussian VAE each satisfy A#1, A#2 and (from §2) A#3, so by Theorem 2 data lie on a manifold with statistically independent sub-manifolds. The VAE defines a similar manifold and its objective is maximised *iff* $p_\theta(x) = p(x)$, hence when distributions over VAE sub-manifolds match those over ground truth sub-manifolds. VAE sub-manifolds map 1-to-1 to latent dimensions by Theorem 2. Identifiability follows from uniqueness of $p_{f,v^*}^{\#(i)}(x)$. $\qquad \square$

### 4.3 NON-LINEAR $f$

Having seen that column orthogonality (A#3) is *sufficient* for independent factors to manifest in $\mathcal{X}$, we consider if it is *necessary*. We relax A#3 and consider a general push-forward distribution under A#1 and A#2 (independent latent variables, $f$ injective) for fully differentiable $f$.

In previous scenarios (linear, column-orthogonal Jacobian), sub-manifolds in $\mathcal{Z}$ ($\mathcal{V}_{v^*}^{(i)}$, $\mathcal{Z}_{z^*}^{(i)}$) are linear, defined by the right singular vectors of the Jacobian, and *constant* $\forall z \in \mathcal{Z}$. Those sub-manifolds can also be defined parametrically, as continuous paths that follow right singular vectors at each point (*cf* integrating over a vector field, see Figure 2 (*right*)).[6] In our now relaxed scenario, singular vectors can vary over $\mathcal{Z}$.

Since the SVD of a matrix $\boldsymbol{M}$ is continuous w.r.t. $\boldsymbol{M}$ (Papadopoulo & Lourakis, 2000), if $\boldsymbol{J}$ is continuous w.r.t. $z$ (by differentiability of $f$) then right singular vectors $\boldsymbol{v}_i$ (as a function of $z$) trace continuous sub-manifolds $\mathcal{V}^{(i)} \subseteq \mathcal{Z}$ that are orthogonal everywhere. By definition of the SVD, mutually orthogonal $\mathcal{V}^{(i)} \subseteq \mathcal{Z}$ map to mutually orthogonal sub-manifolds $\mathcal{M}_f^{(i)} \subseteq \mathcal{M}_f$ (as previously), thus the push-forward distribution over the manifold $\mathcal{M}_f$ again factorises as a product of component push-forward distributions over each $\mathcal{M}_f^{(i)}$. Now, however, sub-manifolds $\mathcal{V}^{(i)} \in \mathcal{Z}$ *need not be linear* and *are not statistically independent* in general, i.e. the density at $z^* \in \mathcal{Z}$ may not factorise as the product of densities over $\mathcal{V}^{(i)} \subseteq \mathcal{Z}$ passing through $z^*$.

Thus, either: (1) sub-manifolds $\mathcal{V}^{(i)} \subseteq \mathcal{Z}$ are not statistically independent and $p(x)$ does not factorise as a product of *independent* components (in simple terms, "$f$ *entangles* $z_i$"); or (2) sub-manifolds $\mathcal{V}^{(i)} \subseteq \mathcal{Z}$ are statistically independent (e.g. $\mathcal{V}^{(i)}$ form an arbitrary orthogonal basis) and are mapped by $f$ to independent components in $\mathcal{X}$. In case (2), since an optimal Gaussian VAE maps independent components $\mathcal{M}^{(i)} \subseteq \mathcal{X}$ to the standard basis in $\mathcal{Z}$, independent factors in $\mathcal{X}$ can be identified, but sub-manifolds $\mathcal{V}^{(i)} \subseteq \mathcal{Z}$ are *unidentifiable*, analogous to $\boldsymbol{V}$ in the linear case. In other words, applying local $\boldsymbol{V}$-basis transformations everywhere (a continuous mapping) can be considered collectively as a *global non-linear basis transformation* that doesn't change the probability distribution, hence is irrecoverable.

## 5 INTERPRETING $\beta$ OF $\beta$-VAE

We now consider the role of $\beta$ parameter in the $\beta$-VAE objective (Eq. 1), which is empirically observed to affect disentanglement (Higgins et al., 2017). Previous works interpret $\beta$ as re-weighting the KL and reconstruction components of the ELBO, or serving as a Lagrange multiplier for a KL "constraint". We provide an interpretation more in keeping with the original ELBO.

To model data from a given domain, a ($\beta$-)VAE requires a suitable likelihood $p_\theta(x|z)$, e.g. a Gaussian likelihood for coloured images, and a Bernoulli for black and white images where pixel values $x^k \in [0, 1]$ are bounded (Higgins et al., 2017) . In the Gaussian case, dividing Eq. 1 by $\beta$ shows that training a $\beta$-VAE with encoder variance $\text{Var}[x|z] = \sigma^2$ is *equivalent to a VAE* with $\text{Var}[x|z] = \beta\sigma^2$ and adjusted learning rate (Lucas et al., 2019). We now interpret $\beta$ for other likelihoods.

In the Bernoulli example mentioned above, black and white image pixels are not strictly black *or* white ($x^k \in \{0, 1\}$) and may lie between ($x^k \in [0, 1]$), hence the Bernoulli distribution appears invalid as it does not sum to 1 over the domain of $x^k$. That is, unless each sample is treated as the mean $\bar{x}$ of multiple (true) Bernoulli samples. Multiplying the likelihood by a factor $\kappa > 1$ is then tantamount to scaling the number of observations as though each were made $\kappa$ times, lowering the variance of the "mean" observation, $\text{Var}[\bar{x}] \overset{\kappa \to \infty}{\longrightarrow} 0$.[7] Thus, multiplying the KL term by $\beta$ in a $\beta$-VAE, or equivalently dividing the likelihood by $\beta$, amounts to scaling the likelihood's variance by $\beta$, just as in the Gaussian case: **higher $\beta$** corresponds to lower $\kappa$ ("fewer observations") and so **higher likelihood variance**. Since the argument holds for any exponential family likelihood, we have proved

**Theorem 3** ($\beta$-VAE$_{\sigma^2} \equiv$ VAE$_{\beta\sigma^2}$)**.** *If the likelihood $p_\theta(x|z)$ is of exponential family form, a $\beta$-VAE with $\text{Var}[x|z] = \sigma^2$ is equivalent to a VAE with $\text{Var}[x|z] = \beta\sigma^2$.*

---

[6]e.g. define path $i$ passing through $z^* \in \mathcal{Z}$ by $\mathcal{V}_{v^*}^{(i)} = \{v^{(i)}(t) \mid v^{(i)}(t) = v^* + \int_0^t \frac{\partial v_i}{\partial z}(v(t))dt\}$, where $\frac{dv}{dt} = \frac{\partial v_i}{\partial z}(v(t)) = \boldsymbol{v}_i$ is right singular vector of the Jacobian evaluated at $v(t)$.

[7]A mode-parameterised Beta distribution could also be considered, but we keep to a more general argument.

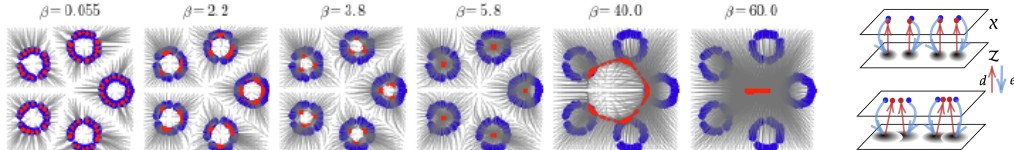

Figure 3: Effect of $\text{Var}[x|z]$, or $\beta$, on reconstruction (blue = data, red = reconstruction): (*l*) For low $\beta$ ($\beta = 0.55$), $\text{Var}[x|z]$ is low, by Eq. 6 & 9, and data are closely reconstructed (see right, top). As $\beta$ increases, $\text{Var}[x|z]$ and so $\text{Var}[z|x]$ increase and posteriors of nearby data points $\{x_i\}$ increasingly overlap (see right, bottom). For $z$ in overlap of $\{q(z|x_i)\}$, the decoder $\mathbb{E}[x|z]$ maps to a weighted average of $\{x_i\}$. Initially, close neighbours map to their mean ($\beta = 2.2, 3.8$), then small circles "become neighbours" and map to their centroids, until finally all samples map to the global centroid ($\beta = 60$). (reproduced with permission from Rezende & Viola, 2018) (*r*) illustrating posterior overlap, (*t*) low $\beta$, (*b*) higher $\beta$.

In the most general case, the $\beta$-ELBO (Eq. 1) is maximised if $q(z|x) \propto p_\theta(x|z)^{1/\beta}p(z)$, and $\beta$ can be interpreted as a *temperature* parameter: high $\beta$ dilates the likelihood towards a uniform distribution (high $\text{Var}[x|z]$), low $\beta$ concentrates it towards a delta distribution (low $\text{Var}[x|z]$).

Figure 3, from Rezende & Viola (2018), nicely illustrates the effect of varying $\beta$ and empirically demonstrates the relationship to $\text{Var}[x|z]$. As variance increases, posteriors of nearby data points $\{x_i\}$ (blue) increasingly overlap (by Eq. 6/9) and the decoder maps latents in regions of overlap to weighted averages of $x_i$ (red). Since $\text{Var}[x|z]$ governs how close data points need to be for this effect, it acts as a "glue" over $x \in \mathcal{X}$ (see caption for details).

In §3, we saw that optimising the ELBO encourages Jacobian orthogonality, on which disentanglement relies, in expectation over posteriors (Eq. 8). We conjecture that this justifies why increased $\beta$ enhances disentanglement (Higgins et al., 2017; Burgess et al., 2018): increasing $\beta$ increases $\text{Var}[x|z]$ and so $\text{Var}[x|z]$ (Eq. 8), which (i) encourages orthogonality over a broader region of $\mathcal{Z}$; and (ii) increases posterior overlap where multiple orthogonality constraints apply simultaneously (Fig. 3).

We note that Theorem 3 also allows clearer interpretation of other works that vary $\beta$. While setting $\beta > 1$ can enhance disentanglement, setting $\beta < 1$ is found to mitigate "posterior collapse" (**PC**), which describes when a VAE's likelihood is sufficiently expressive such that it learns to directly model the data distribution, $p(x|z) = p(x)$, leaving latent variables redundant (Bowman et al., 2015).

**Corollary 3.1** ($\beta < 1$). *Setting $\beta < 1$ is expected to mitigate posterior collapse.*

*Proof.* From Theorem 3, $\beta < 1$ reduces $\text{Var}[x|z]$, constraining the distributional family that $p_\theta(x|z)$ can describe. For some $\beta$, $\text{Var}[x|z] < \text{Var}[x]$ and so $p(x) \neq p_\theta(x|z)$, $\forall \theta$, making PC impossible. $\square$

# 6 RELATED WORK

Many works study aspects or variants of VAEs, or disentanglement in other modelling paradigms. Here, we review those that offer insight into understanding the underlying cause of disentanglement in VAEs. Higgins et al. (2017) first showed that disentanglement is enhanced by increasing $\beta$ in Eq. 1, and Burgess et al. (2018) hypothesised that diagonal posterior covariances may be the cause, encouraging latent dimensions to align with generative factors of the data. Rolinek et al. (2019) empirically showed and theoretically supported a link between diagonal posterior covariances and orthogonality in the decoder Jacobian, deemed responsible for disentanglement. Kumar & Poole (2020) simplified and generalised the argument. These works demonstrate that diagonal posteriors provide an inductive bias that breaks the rotational symmetry of an isometric Gaussian prior, side-stepping impossibility results related to independent component analysis (e.g. Locatello et al., 2019).

Several works investigate analytically tractable linear VAEs (Lucas et al., 2019; Bao et al., 2020; Koehler et al., 2022). Zietlow et al. (2021) show that disentanglement is sensitive to perturbations to the data distribution. Reizinger et al. (2022) relate the VAE objective to *independent mechanism analysis* (Gresele et al., 2021), which encourages column-orthogonality in the mixing function of ICA, similarly to that induced implicitly in the decoder of a VAE. Ramesh et al. (2018) trace independent factors by following leading left singular vectors of the Jacobian of a GAN generator. In the opposite direction, Chadebec & Allassonnière (2022) trace manifolds in latent space by following a locally averaged metric derived from VAE posterior co-variances. Pan et al. (2023) claim that the

data manifold is identifiable from a geometric perspective assuming Jacobian-orthogonality, which differs to our focus on statistical independence. More recently, Bhowal et al. (2024) consider the encoder/decoder dissected into linear and non-linear aspects, loosely resembling our view of the Jacobian in terms of its SVD. However, the decoder function is quite different to its Jacobian and dissecting a function into linear and non-linear components is not well defined whereas an SVD is unique.

Recently, Buchholz et al. (2022) analysed several function classes identifiable by Independent Component Analysis (ICA), including conformal maps. This relates closely to our analysis of a decoder with column-orthogonal Jacobian (§4.2), which is a conformal map (see Def. 2, Buchholz et al., 2022). Conformal maps are proved to be identifiable in abstract via Moebius transforms, whereas we give a constructive proof for VAEs in terms of the SVD of the decoder's Jacobian. Combining these appears a promising direction to better understand the interplay between stochastic and deterministic approaches to learning latent generative factors.

## 7 CONCLUSION

Unsupervised disentanglement of independent factors of the data is a fundamental aim of machine learning and significant recent progress has been made in the case of VAEs. We extend that work by showing: (i) that the previously proposed approximate relationship can be defined precisely; and (ii) that the choice of diagonal posterior covariances in a VAE causes statistically independent components of the data to align with distinct latent variables of the model, i.e. disentanglement. In the process, we provide a novel yet straightforward interpretation of $\beta$ in a $\beta$-VAE, which plausibly explains why increasing $\beta$ promotes disentanglement but degrades generation quality; and why decreasing $\beta$ mitigates posterior collapse. We also supplement the proof of orthogonality by showing that the likelihood's Hessian is necessarily encouraged to be diagonal and giving a detailed analysis of the Jacobian's optimal singular values.

Neural networks are often considered too complex to explain, yet recent advances make their deployment in everyday applications all but inevitable. Improved theoretical understanding is therefore essential to be able to confidently take full advantage of machine learning progress in non-trivial and potentially critical systems, and we believe that the body of work that we add to here is a useful step. Interestingly, our approach rests on the fact that, regardless of the model's complexity, its Jacobian, which transforms the density of the prior, can be considered in relatively simple terms.

Not only is a better understanding of VAEs of interest in itself, VAEs are often part of the pipeline in recent diffusion models that achieve state-of-the-art generative performance (e.g. Pandey et al., 2022; Yang et al., 2023; Zhang et al., 2022). Other recent works show that supervised learning (Dhuliawala et al., 2023) and self-supervised learning (Bizeul et al., 2024) can be viewed from a latent variable perspective and trained under a suitable variant of the ELBO, connecting VAEs to other learning paradigms in a common mathematical language.

One limitation of our work and of current understanding more generally is that disentanglement is observed in VAEs with non-Gaussian likelihoods (Higgins et al., 2017), whereas current work, including ours, focus predominantly on the Gaussian case. We plan to address this in future work.

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

# A   APPENDIX: PROOFS

## A.1   PROOF OF THEOREM 1

We summarise the key logical steps outlined in §4.1 with notation as defined there: $\boldsymbol{D} = \boldsymbol{U}\boldsymbol{S}\boldsymbol{V}^\top$, $\boldsymbol{u}_i, \boldsymbol{v}_i$ are columns of $\boldsymbol{U}, \boldsymbol{V}$ (left/right singular vectors), $s_i = \boldsymbol{S}_{i,i}$ are singular values.

*Proof.*

- we can consider sampling $z \sim p(z)$ as sampling random variables $\mathbf{v}_i$ with realisations defined in the $\boldsymbol{V}$-basis: $v_i = \boldsymbol{v}_i^\top z$;

- since $p(v) = \prod_i p(v_i)$ and $p(v_i) = \mathcal{N}(\mathbf{v}_i; 0, 1)$, $\mathbf{v}_i$ are independent standard normal;

- $p_{f,v}^{\#(i)}(x) = |s_i|^{-1} p(v_i)$ are push-forward distributions w.r.t. $\mathbf{v}_i$ under $f$;

- if we define $u = \boldsymbol{U}^\top x$ ($x$ in the $\boldsymbol{U}$-basis), then $p(x) = p(u)$ and $\boldsymbol{J} = \boldsymbol{U}\boldsymbol{S}\boldsymbol{V}^\top = \frac{\partial x}{\partial u}\frac{\partial u}{\partial v}\frac{\partial v}{\partial z} = \boldsymbol{U}\frac{\partial u}{\partial v}\boldsymbol{V}^\top$, which implies a simple relationship: $\frac{\partial u}{\partial v} = \boldsymbol{S}$, or $\frac{\partial u_i}{\partial v_j} = \{s_i \text{ if } i = j, \text{ o/w } 0\}$;

- thus $p(u) = \prod_i |s_i|^{-1} p(v_i)$ and also $p(u_i) = |s_i|^{-1} p(v_i)$, such that $p(x) = \prod_i p(u_i)$ factorises;

- comparing definitions above, we see that $p(u_i) = p_{f,v}^{\#(i)}(x)$, i.e. the push-forward distribution of $z$ in the $\boldsymbol{V}$-basis under $f$ equates to a distribution over $x$ in the $\boldsymbol{U}$-basis;

- each $u_i$ is a function of only $v_i$, hence $u_i$ (each co-ordinate of $x$ considered in the $\boldsymbol{U}$-basis), can be treated as a realisation of an independent random variable.

Thus, samples $v_i$ of independent random variables in $\mathcal{Z}$, map separably under $f$ to samples $u_i$ of independent random variables in $\mathcal{X}$; and basis vectors $\boldsymbol{u}_i \in \mathcal{X}$ identify independent components.  □

To perhaps illustrate the linear case more clearly, we also know $u = \boldsymbol{S}v$, $u_i = s_i v_i$ and can explicitly define the independent components of $x$: $p(u_i) = \mathcal{N}(u_i; 0, s_i^2)$. It is thus clear that $p(x)$ factorises as a product of these independent factors:

$$
\begin{aligned}
p(x) = \mathcal{N}(x; 0, \boldsymbol{D}\boldsymbol{D}^\top) &= (2\pi)^{-D/2} |\boldsymbol{D}\boldsymbol{D}^\top|^{-1} \exp\left\{ -\tfrac{1}{2} x^\top (\boldsymbol{D}\boldsymbol{D}^\top)^{-1} x \right\} \\
&= (2\pi)^{-D/2} |\boldsymbol{S}^2|^{-1} \exp\left\{ -\tfrac{1}{2} x^\top \boldsymbol{U}\boldsymbol{S}^{-2}\boldsymbol{U}^\top x \right\} \\
&= (2\pi)^{-D/2} \left( \prod_i s_i^{-2} \right) \exp\left\{ -\tfrac{1}{2} u^\top \boldsymbol{S}^{-2} u \right\} \\
&= \left( \prod_i (2\pi)^{-1/2} s_i^{-2} \right) \exp\left\{ -\sum \tfrac{1}{2s_i^2} u_i^2 \right\} \\
&= \prod_i (2\pi)^{-1/2} s_i^{-2} \exp\left\{ -\tfrac{1}{2s_i^2} u_i^2 \right\} \qquad = \prod_i \mathcal{N}(u_i; 0, s_i^2)
\end{aligned}
$$

## A.2   PROOF OF COROLLARY 1.1

*Proof.* Noise parameter $\sigma$ can be learned in PPCA (see Tipping & Bishop, 1999), which we assume is well-approximated or otherwise known ($\sigma_{\text{PPCA}} = \sigma$).

- The PPCA model (Eq. 2) satisfies the assumptions of Theorem 1.

- By Theorem 1, for data generated under the PPCA model, the probability density over mean parameters $\hat{x} = \boldsymbol{D}z$ factorises as $p(\hat{x}) = \prod_i p_{f,v}^{\#(i)}(\hat{x})$ with $p_{f,v}^{\#(i)}(\hat{x}) = |s_i|^{-1} p(v_i) = p(u_i)$, where $v = \boldsymbol{V}^\top z$, $u = \boldsymbol{U}^\top \hat{x}$ and basis vectors $\boldsymbol{u}_i$ correspond to independent components of $\hat{x}$.

- With sufficient data, the ELBO of an LVAE is maximised when $\boldsymbol{D}_{\text{LVAE}} = \boldsymbol{U}\boldsymbol{S}$, hence columns $s_i \boldsymbol{u}_i$ of $\boldsymbol{D}_{\text{LVAE}}$ identify independent components of the data.

- If all $s_i$ are distinct, $\boldsymbol{D}$ is fully identified by its SVD, which is unique up to column permutation and sign.  □

