# OpenReview forum: "Unpicking Data at the Seams: VAEs, Disentanglement and Independent Components"
_ICLR.cc/2025/Conference — Submitted to ICLR 2025_

### Official Review · Reviewer_KWdN · 2024-10-29

**Soundness:** 3
**Presentation:** 2
**Contribution:** 1
**Rating:** 3
**Confidence:** 4

**Summary:**

The paper connects orthogonality of the decoder Jacobian in VAEs to disentangling statistically independent latent factors.

**Strengths:**

The paper is
- generally well-written,
- the figures are high quality,
- the structure is logical and easy to follow.

**Weaknesses:**

Despite the neat structure and the obvious effort the authors put into this work, **I honestly struggled to discern what the added value of the contribution of the paper is, compared to the literature** (while knowing the field and having worked on related projects). I will try to make the points that confused me clear below. Please let me know if my understanding is incorrect.

## Major issues
- The added value of the contribution is unclear to me. A potential statement for this could be the _wrong_ statement, made in L468: _"Reizinger et al. (2022) relate the VAE objective to independent causal mechanisms (Gresele et al., 2021) which consider non-statistically independent sources that contribute to a mixing function by orthogonal columns of the Jacobian. This clearly relates to the orthogonal Jacobian bias of VAEs, but differs to our approach that identifies statistically independent components/sources"_. Namely, **Reizinger et al. (2022) assume a standard VAE with statistically independent sources.** It seems to me that the authors themselves differentiate their work from Reizinger et al. (2022) only by the presumed (but not true) dependent/independent sources divide. As this is not the case (unless my understanding is incorrect), I cannot see the added value of the paper. This holds for the theorems, and also for the elucidation of the role of $\beta,$ as Reizinger et al. (2022) also relate it to the decoder variance in their Appx. A.3.
- The paper is a bit too heavy on notation in the main text. While I understand that the proofs require precision, communicating (the intuition behind) the results do not. Please rework the main text to include only the necessary notation. Also, please always explain what your notation means (e.g., the caption of Figure 1 is not self-contained)

## Minor points
- L167: expand of what you are trying to do with the two losses
- Eq (9): define $s_i$
- L248: why is the image manifold parallel to the $U$-basis?
- Eq (12): it should include the log absolute determinant of teh Jacobian, and not $W$ as this is about the nonlinear case, right?

**Questions:**

- What do the two types of bold D's in Eq. (6) denote?
- L156-160: isn't statistical independence of latent factors assumed in VAEs?

---

> ### Author Response · Authors · 2024-11-15
> **Author Rebuttal**
>
> Thank you once again for your review, we have improved the paper and look forward to any further feedback.
>
> * **Re [1]**: we hope to have addressed your main concern in the general response. Our work offers an intuitive constructive proof of disentanglement starting from the ELBO through to a description of the factor distributions. We draw close analogy to the linear case for intuition. We also now include an *identity* of the key relationship between posterior covariance and log likelihood Hessian, in place of previous Taylor approximations.
>     * As noted above, we will reconsider how to reference [1].
> * **Readability**: thank you for the feedback, we have reworded the main theoretical section (same content) to hopefully be more readable. We have updated the figure captions to summarise the key logial steps.
>
> **Minor**:
> * **L167** - this is now removed as we replace the det-ELBO approximation with the identity
> * **Eq 9** (now 11) - $s_i$ is defined a few lines above as a singular value of the Jacobian.
> * **The $U$-basis** is a basis for $\mathcal{X}$ defined by the right singular vectors of the Jacobian (solid red lines in the image), i.e. of the matrix $D$ in the linear case (since $x=Dz$). This section steps through the simpler linear case in a way that then holds in the non-linear case.
> * **typo**: fixed, thank you.
>
> **Questions**:
> * **Ds**: These are the Jacobian and Hessian of the decoder, desribed beneath the equation. We have adjusted the wording to hopefully improve clarity.
> * **L156-160**: That is correct, but here we are referring to a link betwen orthogonality of partial derivatives (related to linear independence) to disentanglement (related to statistical independence).

---

> > ### Comment · Reviewer_KWdN · 2024-11-19
> >
> > Thank you for the clarifications. I will respond to your answer regarding [1] below the joint response.
> >
> > Please note that Gresele et al., 2021 considers statistically independent sources, i.e., this sentence at line 481 needs to be corrected:
> > > Reizinger et al. (2022) relate the VAE objective to independent causal
> > mechanisms (Gresele et al., 2021) which consider non-statistically independent sources that contribute to a mixing function by orthogonal columns of the Jacobian.

---

> > > ### Author Response · Authors · 2024-11-21
> > >
> > > Thank you. We have updated the reference to Gresele et al, 2021 in the updated manuscript.

---

### Official Review · Reviewer_YxoF · 2024-11-03

**Soundness:** 3
**Presentation:** 2
**Contribution:** 3
**Rating:** 5
**Confidence:** 2

**Summary:**

The paper provides a theoretical understanding of disentanglement within Variational Autoencoders when the columns of the jacobian of the decoder are orthogonal (theorem 2).

Starting from this hypothesis, namely the linear independence of the columns of the decoder Jacobian, the author shows that it is possible to link statistically independent components of the output distribution to independent latent factors of variation.

**Strengths:**

The authors have clearly defined the problem and the goal of the paper;

The analysis extends existing work, offering theoretical proof of why an architectural constraint such as diagonal posterior covariance induces an effective disentanglement in the VAE latent space.

**Weaknesses:**

The paper lacks an experimental part.
Even if the contribution is theoretical, a small experiment on a synthetic dataset corroborates the claims and strengthens the contribution.

Splitting the deductive reasoning between ll 210 to 259 in paragraphs highlighting the key logical steps can also be helpful for the reader.

Additional figures or clarification of the only existing one (Fig 1) can help the reader understand the sequence of logical steps more easily.

**Questions:**

How sensitive is the disentanglement effect to minor violations of the assumptions (orthogonality of the decoder Jacobian)?

in 059: do you mean encoder or decoder Jacobian?

---

> ### Author Response · Authors · 2024-11-15
> **Author rebuttal**
>
> Thank you again for your review, we hope to address your main concerns below and look forward to any further feeback.
>
> * **Experimental part**:
>     * Our paper completes a thread of theoretical works to fully explain disentanglement in VAEs. This result has wide generality since VAEs can be used for many data types and VAEs are combined with other modelling paradigms, e.g. state-of-the-art diffusion models. Many works demonstrate disentanglement (e.g. [5]) and several empirically verify that disentanglement and orthgonal Jacobian go hand-in-hand [3,4], and the effect of beta is demonstrated in [5,7]. We fully agree that empirical results are necessary to validate to theoretical claims in general, and everything that we explain has been empirically verified.
>     * The field of machine learning currently has far more empirical results than theoretical explanations and many papers are published showing new empirical results without firm theoretical basis (which we are not criticising). Given we are explaining well-documented and important phenomenon, fundamental to the aims of machine learning, we hope that it is acceptable for a paper to be published to help redress the imbalance between empirical results and theoretical understanding.
> * **Deductive reasoning**:
>     * Thank you for the feedback, we have updated the wording of the main theoretical section and hope it is now more readable and easier to parse.
> * **Figures**:
>     * We have split the figure so that it is easier to view alongside the main text and also updated the captions so they better summarise the key logical steps.
>
> [7] [Taming VAEs](https://arxiv.org/abs/1810.00597)

---

> > ### Comment · Reviewer_YxoF · 2024-12-02
> >
> > I appreciate the authors' response to my concerns and their effort to clarify their message for the other reviewers.
> >
> > However, as I am not fully familiar with certain parts of the relevant literature on this topic, I can not express strong opinions either in favor of or against this work, as some of my fellow reviewers have done.

---

> ### Author Response · Authors · 2024-12-02
>
> Many thanks for reviewing our work.
>
> We have improved the presentation of our main results as you requested ("Deductive reasoning"), and hope that these are now easier to follow. These step through how a diagonal covariance matrix causes the manifold distribution to factorise into independent components that are axis-aligned directions in latent space -- a mathematical way to describe "disentanglement". **If this is now more clear, we ask that you might consider increasing your score.**
>
> Separately, if, as you say, you are less familiar with parts of the literature, it would be **very helpful if you could independently confirm [the issues we raise with [1]](https://openreview.net/forum?id=HuL2yba6Uf&noteId=Bg0xlgg5TV)**. We show 2 simple cases where textbook inequalities are incorrectly applied, which are immediate to verify, invalidating the proof. This would help clarify to all/AC that our work ought not to be compared to [1] (which we anticipate being retracted).
>
> Many thanks,
>
> Paper Authors
>
> [1] [Embrace the Gap: VAEs Perform Independent Mechanism Analysis (2022)](https://arxiv.org/abs/2206.02416)

---

### Official Review · Reviewer_fhCg · 2024-11-04

**Soundness:** 2
**Presentation:** 3
**Contribution:** 1
**Rating:** 3
**Confidence:** 3

**Summary:**

This paper aims to make progress on understanding why VAEs are able to learn disentangled representations. To this end, the authors connect orthogonality of the decoder Jacobian to disentanglement by showing that the VAE learns statistically independent sub-manifolds in the observed space. Further, the paper provides additional insights into the VAE objective such as on the role of beta in a beta-VAE.

**Strengths:**

* The paper addresses an important problem. Namely, developing a principled understanding of how and why disentanglement is possible in deep generative models.

* The paper is well written and the presentation is well structured.

* The theoretical results and insights are presented in an intuitive and accessible way, with nice visual intuition from the figures.

**Weaknesses:**

My main issue with this paper is that I do not think the contributions presented in this paper offer sufficient novelty relative to prior work to merit acceptance, and, moreover, I do not think the authors adequately compare their contribution to prior work. I discuss these points in detail below with specific examples.

**Identifiability of Latent Factors in VAEs**

The authors build on several prior works which try to explain why VAEs are able to identify the ground-truth latent factors by showing that the VAE objective promotes the decoder to have an orthogonal Jacobian. The authors perform a similar analysis and present a result claiming to be an identifiability result for VAEs.

Understanding the identifiability of VAEs, however, has been analyzed in detail in prior works. Specifically, the work of [1], rigorously showed that the VAE objective with vanishing decoder variance is equivalent to maximum likelihood under an independent Gaussian prior plus a regularization term enforcing that the Jacobian has orthogonal columns. The identifiability of such models with independent latents and orthogonal Jacobians was studied in depth by [2], who proved a certain form of identifiability for such models and showed the theoretical challenges in recovering a complete identifiability result.

The authors do not mention the work of [2] and wrt [1], they state that their analysis differs because [1] assumes statistically dependent factors such that there result is novel relative to this work. This statement is wrong. The work in [1] does not assume statistically dependent factors. To this end, I think the works of [1] and [2] conduct a more rigorous and comprehensive analysis of the VAE objective and its identifiability than the current work, such that I do not feel this works adds sufficient novelty.

Additionally, the analysis of the VAE objective and the identifiability analysis conducted in this work are not as rigorous as these prior works in [1, 2], and are closer to the results in [3, 4]. From this standpoint, however, I am also not sure what the novelty is of the authors SVD based identifiability argument, as similar ideas were presented in [3] as I understand. Perhaps, the authors can clarify this as well.

**Understanding the Role of Beta in Beta-VAEs**

Another stated contribution of this work is the authors' analysis of the role of beta in a beta-VAE (Section 3.3). The authors state that beta can be interpreted "as scaling the variance of the likelihood distribution,". From what I can tell, this result also seems very similar to the result in [1] presented in Appendix A.3 on the role of beta. I am curious if the authors can comment on the novelty of their result relative to this prior work.


**Theorem Statements**

As an additional point, I did not see proofs for Theorems 1 and 3, and am thus curious if I am missing something or if there are proofs for these results.

**Bibliography**

1. Embrace the Gap: VAEs Perform Independent Mechanism Analysis
 (https://arxiv.org/abs/2206.02416)

2. Function Classes for Identifiable Nonlinear Independent Component Analysis
(https://arxiv.org/abs/2208.06406)

3. Variational Autoencoders Pursue PCA Directions (by Accident)
(https://arxiv.org/abs/1812.06775)

4. On Implicit Regularization in β-VAEs (https://arxiv.org/abs/2002.00041)

**Questions:**

**1.** Can the authors comment on the novelty of their results relative to the prior results discussed above?

**2.** Where are the proofs for Theorems 1 and 3?

---

> ### Author Response · Authors · 2024-11-15
> **Author rebuttal**
>
> Thank you again for your review and in particular for raising reference [6] (your [2]), of which we were unaware.
>
> * [6]: this work is very interesting, relevant and complementary to our own and we have made reference to it. As now noted in our paper, [6] provides a proof of identifiability of abstract conformal maps based on Moebius tranforms, whereas we provide a constructive proof based on the SVD of the Jacobian with analogy to the linear case for insight. It will be of interest to further consider the relationship between these works. Many mathematical results are proven in many ways and the relationships between proofs can offer new insights, so we hope that concerns of overlap or differences between these approaches are resolved.
> * Differences to prior works [1,2,3] are addressed in the note to all reviewers.
> * Proofs of Theorems 1 and 3 are given before the theorem is stated. For the purposes of narrative, we develop the argument and conclude with what has been proven.
>
>
> [6] [Function Classes for Identifiable Nonlinear Independent Component Analysis](https://arxiv.org/abs/2208.06406)

---

> > ### Comment · Reviewer_fhCg · 2024-11-26
> >
> > I thank the authors for their reply and changes made to the manuscript. I continue to share the sentiment towards this work expressed in my initial review, however.
> >
> > Specifically, I still do not think the authors contextualize their contributions adequately relative to existing theoretical work **on VAEs**. Regarding the work from [1], the authors initially made a false claim regarding their relationship with this work, and now claim that the work in [1] should be disregarded because of purported errors in their proof. I find this line of reasoning unconvincing. If the authors would like to claim that the results in [1] are wrong and will not hold, then this should be done more rigorously and should have been a central point of their current work upon its initial submission.
> >
> > Furthermore, the authors continue to make claims regarding the work from [1] such as “The key mathematical arguments in [1] mirror those in [2]” which, to the best of my knowledge, are not true. My understanding is that the mathematical results in [1] differs significantly from [2] in that [1] starts with the VAE objective, and rigorously takes the limit as the decoder variance goes to zero, while [2] relies on a more ad hoc Taylor expansion w.r.t. the encoder variance.
> >
> >
> > I also still do not think the authors contextualize their contributions adequately relative to existing work on **identifiability**. My understanding is that the authors are essentially claiming to have an identifiability result for latent variable models with independent latents and orthogonal decoder Jacobian. If this were true, it would be quite significant, as substantial work has gone into arriving at such a result in the identifiability community [3]. It is difficult to understand if this is the case, however, as the author's current result does not possess the same level of rigour as existing identifiability results (e.g. see [3, 4, 5, 6]). For example, the authors do not ever even give a formal definition of what they mean by “identifiability”.
> >
> > Thus, it remains difficult to understand whether the present work offers any new contribution to either of these communities based on the current state of the paper. I am thus maintaining my score and recommend rejection. For future iterations of this work, I would encourage the authors to better position their contributions within the existing identifiability or VAE literature.
> >
> > [1] Reizinger et. al, 2022 Embrace the Gap: VAEs Perform Independent Mechanism Analysis
> >
> > [2] Kumar et. al, 2022 On implicit regularization in $\beta$-VAEs
> >
> > [3] Buchholz et. al, 2022 Function classes for identifiable nonlinear independent component analysis
> >
> > [4] Gresele et. al, 2021 Independent mechanism analysis, a new concept?
> >
> > [5] Khemakhem et. 2020 Variational Autoencoders and Nonlinear ICA: A Unifying Framework
> >
> > [6] Lachapelle et. al, 2023 Additive decoders for latent variables identification and cartesian-product extrapolation

---

### Author Response · Authors · 2024-11-15
**Author Response to All**

Dear reviewers, many thanks for your time and detailed consideration of our work. We believe we can address all concerns raised and look forward to discussing any that remain. While we hope the scores will improve in due course, we are delighted that our work is considered *well written/clearly set out* (all), *accessible/easy to follow* (fhCg, KWdN) and *addresses an important problem* (fhCg). We reply separately to each reviewer in detail and provide here a general response to all reviewers on key points.

1) **Prior work [1]**: Reviewers **fhCg** and **KWdN** raise similar concerns regarding novelty vs [1]. We chose not to critique this work in our paper, but we must now address it.
    * [1] appears to contain **material errors** in the proof of Theorem 1 (their main result), e.g.:
        * (p.33) the reverse triangle inequality is used with *square norms*, which [does not hold in general](https://math.stackexchange.com/questions/1671911/squared-reverse-triangle-inequality)
        * (p.33) the penultimate inequality drops unbounded cross terms without explanation
        * (p.34) without explanation, the 1st equation drops the K term (from p.33) and thus all second and higher order derivatives, which are unbounded.
    * [1] makes **strong assumptions** that do not hold in general and are not made in our work or similar works [2,3], e.g.
        * the latent space and data space must have the same dimension
        * Var$[x|z] \to 0$, whereas disentanglement is seen to improve as variance (i.e. $\beta$) *increases* [5]
    * The key mathematical arguments in [1] mirror those in [3] (an earlier work, summarised in our paper) and it is unclear that the differences are valid (as above)

    We will reconsider how [1] is referenced in our work but hope this sufficiently addresses the concerns raised.

2) **Novelty**: independently of the above, we emphasise the novelty of our work (note that to explain disentanglement from the ELBO requires steps (**A**)  from ELBO to orthogonality; and (**B**) from orthogonality to disentanglement. [1,2,3] largely address A, we address B.):
    * we provide a first *constructive* proof of how disentanglement follows from column-orthgonality, with insight by analogy to the linear case (**B**)
    * we generalise the connection between $\beta$ and Var$[x|z]$ for Gaussian VAEs [4] to *exponential family* VAEs, for which disentanglement is observed [5].
    * to further increase novelty over prior work, we include the *precise* relationship (new Eq 8) between posterior covariance and log-likelihood Hessian, previously approximated [1,2,3] (**A**).
    * In summary, to the best of our knowledge we now provide the most complete constructive proof of **A** and **B**

3) **Changes to the paper**:
    * we have made minor or stylistic edits throughout based on feedback (**YxoF0**, **KWdN**), in particular we hope the main theoretical section is now less terse and easier to follow. We agree that this is important to aid the reader.
    * new aspects are highlighted in teal.
    * to simplify and make space for adjustments we have removed Section 4, which was incidental to the main contribution.


[1] [Embrace the Gap: VAEs Perform Independent Mechanism Analysis (2022)](https://arxiv.org/abs/2206.02416)

[2] [Variational Autoencoders Pursue PCA Directions (by Accident) (2019)](https://arxiv.org/abs/1812.06775)

[3] [On Implicit Regularization in β-VAEs (2020)](https://arxiv.org/abs/2002.00041)

[4] [Don’t Blame the ELBO! A Linear VAE Perspective on Posterior Collapse (2019)](https://arxiv.org/abs/1911.02469)

[5] [$\beta$-VAE: Learning Basic Visual Concepts with a Constrained Variational Framework](https://www.cs.toronto.edu/~bonner/courses/2022s/csc2547/papers/generative/disentangled-representations/beta-vae,-higgins,-iclr2017.pdf)

---

> ### Author Response · Authors · 2024-11-18
> **Addendum: Proof summaries and additional diagram**
>
> Further to our response above, we have reflected further and made the following modifications in response to reviewer comments:
> * succinctly summarised the proof of Theorem 1 in Appendix A1 (**fhCg**, **YxoF**, **KWdN**)
> * succinctly stated the proof of Corollary 1 in more formal terms in Appendix A2 (**YxoF**, **KWdN**)
> * include Fig 2 (right) to help illustrate section 4.3, when column-orthogonality of the Jacobian is no longer assumed (**YxoF**).
>
> For ease of reference, we highlight material changes in brown.
>
> We very much hope this makes the paper more digestible and we look forward to any further clarifications.

---

> ### Comment · Reviewer_KWdN · 2024-11-19
>
> Thank you for your detailed response to address our concerns regarding prior work and novelty.
>
>
> ### Prior work
> - **Reverse triangle inequality:** if I am not mistaken: from $| \Vert a \Vert - \Vert b \Vert | \leq \Vert a-b \Vert$, we can conclude the following for the squared norm case:
>   - $| \Vert a \Vert - \Vert b \Vert |^2 =  | \Vert a \Vert - \Vert b \Vert | \cdot | \Vert a \Vert - \Vert b \Vert |$
>   -  where we can bound each term of the right hand side by $\Vert a-b \Vert$,
>   - thus leading to a correct conclusion in [1]
>   - Am I missing something?
>
> ### Strong assumptions
> - **Dimensionality:** I agree that these assumptions are strong, though these are general in ICA theory, and as IMA was formulated for this case, I think it is reasonable to make the theoretical connection.
>     - The authors of [1] show in their Figure 5 that the same phenomenon happens, providing empirical evidence suggesting that the theory holds in a more general case.
>     - Indeed, later work extended IMA to the setting of lower-dimensional latent factors [(Ghosh et al., 2023)](https://openreview.net/forum?id=PbPLk5OcAK)
> - **Variance:**
>   - Please note that [3] also considers a $\beta\to 0$ setting. Also note that [1] acknowledges in the discussion that _"Whether this theoretical regime [i.e., $\beta\to 0$] matches common practice remains an open question"_
>   - What is your explanation that the experiments in [1] back up their claim in their experimental settings that $\gamma^2\to \infty? yields better disentanglement results?
> ### Novelty:
>   - As [1] also connects the orthogonality to identifiability theory. Maybe I am misunderstanding what you mean by _"first constructive proof"? My understanding of [1] is that it also accomplishes both **A** and **B**.
>  - Thanks, I missed the exponential family VAE point. As [1] also has more general results than Gaussians - their Assumption 1 admits a broader family of priors. Could you please compare how much it differs from your result?
>
>
> Based on these concerns, I am keeping my score.

---

> ### Author Response · Authors · 2024-11-19
>
> Thank you for your reply.
>
> **We detail below the errors in [1], which one would typically expect to make a paper an invalid basis for another not to be published.** Further, [1] proposes a theoretical explanation **under strong assumptions that our work doesn't make**, which would again typically mean our work is not precluded from being published. We hope this resolves comparison to [1]. To differentiate our paper positively, note that our approach differs substantially: the stated "main result (Thm. 1)" of [1] (correctness aside) is an approximation, which we supersede with a precise identity as **the starting point** of our novel explanation by which orthogonality leads to disentanglement via the SVD of the decoder Jacobian.
>
> **Novelty**: Re $\beta$, we generalise previous understanding of $\beta$ to exponential family $p(x|z)$ (as used in [5]). [1], Appx A3 specifically makes "the assumption of a Gaussian decoder". The prior is an independent aspect of the model, $p(x|z)p(z)$, so is an orthogonal consideration.
>
> **Material Errors in [1]**: We are happy to be corrected, but we are fairly sure of the following 3 errors - any of which render the conclusion (Theorem 1, the paper's main result) unsafe:
> 1. **[p.33, after "triangle inequality..."]**: &nbsp; &nbsp;   $|\mathbb{E}[$ $\lVert a\lVert^2 - \lVert b\lVert^2$ $]|$ $\leq$ $ \mathbb{E}[$ $\lVert a - b\lVert^2$ $]$
> &nbsp; &nbsp; &nbsp; [where $a = x-f$; $b = -\sum\tfrac{\partial f}{\partial z_k}...]$
>    *  (dropping expectations for clarity) this has the form &nbsp;
>           " $| \lVert a\lVert^2 - \lVert b\lVert^2 | \leq \lVert a - b \lVert^2$ " &nbsp; &nbsp; &nbsp; &nbsp;(__*__)
>    * triangular inequality:  &nbsp; &nbsp; &nbsp;
>            $| \lVert a\lVert - \lVert b\lVert | \leq \lVert a - b \lVert$ &nbsp; &nbsp; &nbsp;
>            $\Rightarrow$&nbsp; &nbsp;
>            $| \lVert a\lVert - \lVert b\lVert |^2 \leq \lVert a - b \lVert^2$  &nbsp; &nbsp; (by squaring, as you say)
>       * this is not the same as (*) as norms are *squared inside the absolute operator* on the LHS.
>    * counter example to (*): &nbsp; &nbsp; let $b = x>0$, &nbsp; $a = x+1$
>       * $\Rightarrow$ &nbsp; &nbsp;
>            $| \lVert a\lVert^2 - \lVert b\lVert^2 | = | 2x + 1 | > 1 = \lVert a - b \lVert^2$, **contradicting** (*)
> 2. **[next step, p.33]**: &nbsp; $ \mathbb{E}[$ $\lVert (c - e) - (d-e)\lVert^2$ $]$ $\leq$
>           $ \mathbb{E}[$ $\lVert c - e\lVert^2$ + $\lVert d - e\lVert^2$ $]$
>           &nbsp; &nbsp; &nbsp; &nbsp; [$c = x$, $d = f(z)-\sum\tfrac{\partial f}{\partial z_k}...)$, $e = f(\mu)]$
>    * this has the form of the standard triangular inequality
>           $ \lVert a - b \lVert \leq  \lVert a\lVert + \lVert b\lVert |$  &nbsp;
>    * but, as above, all norms are squared, expanding the LHS gives an additional cross term, without which the inequality **does not hold** in general
>         * the inequality may still hold if the cross term is bounded, but this is not claimed in the paper
> 3. **[first step, p.34]**: drops the $K$ term, which bounds the decoder's Hessian and higher derivatives (in earlier Taylor expansion)
>     * this equates to a step in the work of Poole & Kumar, but *is not justified (e.g. in Assumption 1)*
>     * since K is unbounded, any conclusions omitting it are **invalid** in general
>     * in particular the identity we state defines the relationship between posterior covariance and the decoder's Hessian, *proving* it cannot be dropped in general.
>
> As above, we hope that this resolves comparison to [1]. We regret having to highlight issues with this work and trust that our comments are taken at face value and our work considered in good faith. We look forward to addressing any further clarifications on our paper.

---

> > ### Comment · Reviewer_KWdN · 2024-11-23
> >
> > Thank you for your clarifications, and sorry for the delayed response. I wanted to ensure I carefully checked your submission and the references again. I think I now have a better understanding of your work. In light of this, these are my remarks and remaining questions:
> >
> > ### Response to technical points in the proofs in [1]
> > 1. Thank you for showing me a counterexample for the triangle inequality and pinpointing my misunderstanding. I understand that there is a case where this operation is false, though this does not imply that this is the case in [1].
> > 2. Same applies here, as you write.
> > 3. The $K$ term is not dropped, but absorbed into $O_{\gamma^2\to\infty}(1/\gamma^2)$, which relies on the self-consistency assumption, (45) in [1], where the bound comes from Lemma 3, i.e., $K$ is bounded, which results from both encoder and decoder being in $C^2$ (diffeomorphisms) with **bounded first- and second-order derivatives**, included as Assumption 1 (iv)
> > 4. Note that as [1] assumes encoder and decoder with bounded derivatives, their result is **not an approximation** in the sense that they do not only use a few terms in the Taylor approximation. They bound the multivariate Taylor series.
> >
> > Also note that Figure 2 in [1] demonstrates that the authors' claims' prediction do hold under their assumptions.
> >
> >
> > ### Assumption strength
> > I have provided a few additional references in my original response to start a discussion about the strength of the two assumptions you refer to. I would appreciate your response to them.
> > Regarding $\beta$, I am not sure that [5] provides sufficient evidence that larger $\beta$ *on its own* is required for disentanglement - as you own result shows, there is an ``equivalence" between $\beta$ and $\sigma^2$, thus, unless the latter is monitored (which I could not find in [5]), I would not make this conclusion (note: I am not saying this is false, I am saying there is not enough evidence I know of).
> >
> >
> > ### Novelty
> >
> > **Extension to $\beta$**
> > Thank you for clarifying that your result is about the decoder, and not the prior. Please make this clear in 1)the contribution list, 2) title of section 5, 3) name of Thm. 3
> > - Your corresponding result, Cor 3.1, relies on a statement about the role of $\beta$ in mitigating posterior collapse. What is the evidence/reference for this statement?
> >
> > **Column orthogonality as assumption**
> > As far as I understand, the prior works you cited *show* that the decoder Jacobian's columns will become orthogonal, whereas you assume this fact. I think it could be useful for the reader to make this clear.
> >
> > **Linear VAE result**
> > For the linear result, I am unsure about whether you _only_ use it as intuition (you start the section with this; my question is about whether this is the _single_ purpose). I would like to be careful to better determine the novelty of this part; comparing it to Prop. 9 in [1] could also be helpful.
> >
> > ### Questions
> > - Eq (8) relies on a Laplace approximation: how is your claim then not requiring approximations?
> > - Typo in L457? ($Var[x|z]$ is repeated twice)
> > - It is unclear to me what we gain from having a factorizing data distribution; could you please explain this? Or are you only using this as an intermediate fact to be able to say something about the latent factors?
> > - I repeat my question and kindly ask you to clarify what you mean by the _"first constructive proof"_?
> >
> > ### Remarks
> > - L482: the mixing function in Gresele et al., 2021 is by definition column-orthogonal, their proposed loss induces this explicitly (not implicitly) in the inference model (the trained neural network encoder; and not the mixing function)
> > - Line 158: Eq (8) contains the Hessian, which you connect to the decoder Jacobian. Please clarify how this is done, as this is probably not obvious to the reader (I know the similar argument in [3], maybe you could add, "similar to [3]"?).
> > - Contributions list has missing references (question marks are showing up)
> > - Line 139: I find that using $D$ for the decoder Hessian is confusing
> > - Eq. (9): where does the expression for the VAE come from? I suspect from  (8), so please refer to that - even then, it is not trivial to have $D^{\top}D$ instead of $H$. Please elaborate (again, referring to [3] would be sufficient for me).

---

> ### Author Response · Authors · 2024-11-25
>
> Thank you for responding, we respond to each query below.
>
> We hope that are responses make clear the novelty of our work over previous. In particular, **no prior work** (that we are aware of) **reaches the conclusions in Theorems 1-3** or shows that **disentanglement equates to factorising the distribution** over the mean manifold.
>
> ---
>
> **Key points**:
> * **Correctness of [1]**:
>   - as a fundamental of mathematics, *if a proof step does not hold, then the overall proof does not hold*. We have shown this to be the case for 2 steps in the proof of Thm 1 (above, e.g. by counter-example), as the reviewer acknowledges. As such, **Thm 1 in [1] is not proven to hold** under the assumptions stated and we would expect it to be accepted that **[1] is not a reasonable comparison to draw** and so comment no further on it.
>   - we also reiterate that **our Eq 8 makes *precise* the approximate relationship in Thm 1** (we say approximate due to O notation, which is not in Eq 8)
> * **factorising the data distribution *is* disentanglement**. We show that independent factors of p(z) are pushed forward to independent factors of p(x). This is our main contribution, described in Theorems 1 & 2  and Eqs 14 & 17
>    - e.g. if features are independent,    p(x = {big, blue, square}) = p(big) p(blue) p(square)
> * __"first constructive$^*$ proof"__: we not only show that the data distribution factorises into independent components, but also what those components are, i.e. push-forward distributions of 1-D independent components in Z.
>    - \* standard mathematical notation: proofs can be non-constructive (e.g. *existence* of Nash Equilibria but without providing an example of one), or constructive, as here
>
> **Other points**
> * **Assumptions in [1]**: as above, we do not comment on assumptions in [1] beyond restating that we do not make them.
> * **$\beta$ in posterior collapse (PC)** - to clarify:
>    - PC occurs if the likelihood can learn the full data distribution, i.e. $p(x|z)=p(x)$ (hence $Var[x|z] = Var[x]$ **(*)**)
>       - as seen for text where $p(x)$ and $p(x|z)$ are both multinomial distributions (e.g. see Bowman et al, 2015, cited immediately above Cor 3.1).
>    - Reducing $\beta$ is found to mitigate PC in practice.
>    - By Thm 3, reducing $\beta$ reduces $Var[x|z]$, thus for some $\beta$, $Var[x|z]$ drops below $Var[x]$ (a constant) and $p(x|z)$ cannot match the variance of $p(x)$ hence PC is impossible by (*).
> * **Column orthogonality**: This is not assumed per se. In §3 we include a precise identity (Eq 8) that justifies why diagonal posterior covariances implies Jacobian column-orthogonality (in expectation), observed empirically. Our main result shows why this orthogonality implies disentanglement (orthogonality in introduced in Assumption A3).
> * **Linear example**. correct, we first present the main argument in the linear case for intuition in the non-linear case. (In itself, that p(x) factorises as a product of distributions over 1-D sub-spaces is well known since p(x) is Gaussian.)
>
> **Questions**
> * Variational inference approximates the posterior by $q(z|x)$ (assumed Gaussian in  VAEs). Laplace approximations approximate a distribution by a Gaussian. Opper &  Archambeau, 2009 show that VI with Gaussian $q(z|x)$ is equivalent to an averaged Laplace approximation. So this is not an assumption we make, rather a way to interpret the common VAE assumption.
> * Typo - fixed
>
> **Remarks**
> * L482: ".. induced implicitly in the decoder of a VAE." - "implicitly" refers to the VAE
> * link from Hessian to Jacobian: clarified.
> * Contribution list: fixed.
> * Notation: we use D throughout to refer to decoder derivatives (following ICLR notation guidelines), particularly as several Hessians are referenced, but we will review the notation for clarity.
> * Eq 9 derives from Eq 8 assuming a Gaussian likelihood, which we have made clear.

---

> > ### Comment · Reviewer_KWdN · 2024-11-25
> >
> > Thank you for your response! For some reason, I cannot see the fixes in the pdf on openreview - is this an issue on my side?
> >
> > - **Approximations**: you write in your review _we include the precise relationship (new Eq 8) between posterior covariance and log-likelihood Hessian, previously approximated,_ which suggested to me that your (8) is not necessarily an approximation. Thanks for clarifying this.
> > - **Disentanglement:** based on my understanding, disentanglement is not defined in data space, but in latent space (humans also tend to think in terms of factors of variation, which are abstractions). This is why I struggle to see why your statement _"no prior work (that we are aware of) reaches the conclusions in Theorems 1-3 or shows that disentanglement equates to factorising the distribution over the mean manifold."_ implies a meaningful conclusion.
> > - **Constructive proof:** thank you. I wonder what this construction actually brings, could you please elaborate?
> > - **Assumptions in [1]*: please do elaborate on this. The statement "we do not assume X" is not very informative without knowing how strong/realistic assumption X is.

---

> ### Author Response · Authors · 2024-11-28
>
> We address related comments from reviewers [**KWdN**](https://openreview.net/forum?id=HuL2yba6Uf&noteId=LMcV3PqQwq) & [**fhCg**](https://openreview.net/forum?id=HuL2yba6Uf&noteId=H8oUc6t0ig). In particular,
>  - we aim to resolve comparison to reference [1]; and
>  - we are pleased that **fhCg** notes our findings are "**significant if true**" and believe our proofs, which involve no complex steps (and are now summarised in the appendix), confirm our claims.
>
> ### Main points
> * **Reference [1]**
>    - **incorrectness**: we kindly ask **KWdN** and **fhCg** to please fully address our **proof that [1] is incorrect** in 3 places, the first 2 in misapplying the (reverse) triangle inequality [(see Prior Work)](https://openreview.net/forum?id=HuL2yba6Uf&noteId=Bg0xlgg5TV). We would anticipate this as grounds for retraction and so comment no further on its assumptions or relationship to other works.
>    - **mentioning incorrectness of [1] in our work** (**fhCg**): we do not detail issues with [1] in our work as our work does not rely on [1] being incorrect. As stated above [(see Novelty)](https://openreview.net/forum?id=HuL2yba6Uf&noteId=jgh8aUAM69), *even if it were correct*, the main result of [1], Thm 1, proposes an *approximate* relationship (due to $O$-notation), that we state *precisely* by an established identity (our Eq 8). Eq 8 is then the *starting point* for our main result: a constructive proof of how disentanglement follows. We trust that this clarifies the significant difference between our work and [1], independent of correctness; and, having shown that [1] is both **provably incorrect *and* does not overlap with our main**, resolves the comparison.
> * **Identifiability** (**fhCg**) - identifiability relates to the uniqueness of a solution up to certain symmetries, or other conditions. Our main focus is not on identifiability so we make no formal definition, rather uniqueness of solutions follows mainly from that of the SVD (column permutation and sign) on which proofs rely.
>
> ### Clarifications to **KWdN**
> * __Equation 8__: We confirm that, as written, Eq 8 is an equality that states the precise relationship that prior works derive an *approximation* to [[e.g. 1, 2]](https://openreview.net/forum?id=HuL2yba6Uf&noteId=H8oUc6t0ig).
> * __Disentanglement definition__: As we note in §2 (Disentanglement), disentanglement is not a "well-defined" mathematical concept, more a phenomenon observed when generated samples change by a single feature as one latent variable is varied. Thms 1/2 & Eqs 14/17 give a mathematical description of disentanglement as the distribution over the manifold factorising into statistically independent components that are functions of independent latent variable aligned with the standard basis.
> * __Constructive proof__: constructive proofs generally provide clearer intuition. Here, independent components in latent space map to independent components in data space, which explains the mechanism behind how changing a single latent variable changes a single independent component/feature of the data.
> * (__"Unseen fixes"__: please elaborate on what fixes you believe do not appear.)

---

> > ### Comment · Reviewer_KWdN · 2024-11-29
> >
> > Thank you for your response. I mentioned above which changes I do not see (and also, e.g., the missing references, see [here](https://openreview.net/forum?id=HuL2yba6Uf&noteId=qawZYs6DTm))

---

> > > ### Author Response · Authors · 2024-11-29
> > >
> > > **Fixes**: apologies, typo and cross-references are fixed and we thought they were in the last updated manuscript, but there was a minor version-control issue. They will appear in the camera ready.

---

> ### Author Response · Authors · 2024-12-02
>
> Dear Reviewers fhCg & KWdN:
> * over the review process you have each drawn multiple comparisons between our work and [1].
> * we have demonstrated fundamental differences and, moreover, that [1] contains [clear errors in its main result](https://openreview.net/forum?id=HuL2yba6Uf&noteId=Bg0xlgg5TV), which is unfortunate but we would expect to lead to its retraction.
> * we hoped this would swiftly resolve matters given the reviewers knowledge of [1] and we are surprised to have not had confirmation on this most fundamental point.
>
> Having addressed all concerns raised, we ask in the time that remains that the errors we identify in [1] (as published) are confirmed and that our work is reviewed on its own merit without comparison to [1].
>
>
>
>
> [1] [Embrace the Gap: VAEs Perform Independent Mechanism Analysis (2022)](https://arxiv.org/abs/2206.02416)

---

### Author Response · Authors · 2024-11-21
**Discussion Phase ending**

Dear Reviewers, we write as the discussion phase is shortly due to end (26/11).

We are grateful for your feedback, which has helped improve the paper. We hope to have now addressed all points raised and kindly ask if you could confirm, or to let us know of anything further we can clarify.  As perhaps the key points, we hope to have:
* clarified the significant differences and resolved any concerns regarding [1] (**fhCg**, **KWdN**) ; and
* sufficiently justified why we do not provide further empirical support given the extensive amount present in the literature. (Related works evidence Jacobian-orthogonality and we effectively solidify/fill gaps in previous theoretical justifications) (**YxoF**)

Our work justifies disentanglement in ($\beta$-)VAEs, a highly intriguing, domain-spanning phenomenon, known of for almost a decade yet still very much of interest. We also give mathematical meaning to $\beta$ and justify why it mitigates posterior collapse (in *non-Gaussian* settings). We believe our work improves theoretical understanding and could be of interest to many in the ML community. We very much hope that, given the updates made over the course of the review period, you might now reconsider it together with your scores.

Many thanks, Paper Authors

[1] [Embrace the Gap: VAEs Perform Independent Mechanism Analysis (2022)](https://arxiv.org/abs/2206.02416)

---

### Meta-Review · Area_Chair_Hfc6 · 2024-12-20

**Metareview:**

This paper analyzes theoretically why variational autoencoders tend to "disentangle" the data in their latent representation, encoding statistically independent factors.  The authors show that the training promotes column orthogonality in the decoder's Jacobian.  They then go on to show that this translates to statistical independence.  The reviewers overall found the paper well written, technically correct and convincing.  One reviewer noted that the figures are "super high quality".  Note, that of the three reviews received, one seems low confidence.  However, the reviewers voiced concerns about lack of experiments, some clarity (too heavy notation) and primarily novelty.  Specifically, the reviewers pointed out that the paper has a similar narrative to [1] (Embrace the Gap: VAEs Perform Independent Mechanism Analysis (https://arxiv.org/abs/2206.02416)), which also has the column orthogonality result demonstrating disentanglement.  There was significant discussion regarding this connection between the reviewers and the authors, and the AC that led to disagreement.  I will elaborate below.

In the related work, this paper [1] was cited but only very briefly: "Reizinger et al. (2022) relate the VAE objective to independent mechanism analysis (Gresele et al., 2021)".  I would agree that this is not properly attributing prior highly-related work.  The authors subsequently added "which encourages column-orthogonality in the mixing function of ICA, similarly to that induced implicitly in the decoder of a VAE.", but the reviewers contend that this mischaracterizes the work.  A primary argument from the authors is that [1] has a technical flaw in a central theorem that renders the paper incorrect, and thus should not be be cited at all.  The reviewers do not seem to agree with this assessment, i.e. regarding correctness or whether the paper is discussed sufficiently.

While I sympathize with the authors, I disagree with their argument regarding novelty.  Even if there was a technical flaw in a major proof (not suggesting this is true, but hypothetically speaking...), one should not pretend as if the paper didn't exist.  It sets a precedent in ideas, methods, etc. that should be discussed constructively.  The papers are so similar, in motivation, claims and end result that there must be an in-depth discussion *in the paper* of what this work adds, or does better.

The reviewers all voted to reject the paper.  After careful deliberation, discussion with the reviewers and input from the authors, I do not believe it would be appropriate to override the reviews.  Therefore I would recommend reject.

In addition to more detailed discussion of and comparison to previous literature, I asked the reviewers what the authors' could do to change their recommendation to an accept.  I'll paste that below (edited) and hopefully it will be helpful to the authors to improve the manuscript for a future submission.

- Elaborating on the factorization in data space, what that means, and how realistic that is (complex data distributions from the real world do not have this property?) - including experimental validation of this claim
- The authors refer to independent submanifolds, but (statistical) independence is defined for random variables, so this needs a formal treatment.
- An in-depth discussion of the literature, particularly the work by Buchholz et. al, 2022, titled "Function classes for identifiable nonlinear independent component analysis"
- Correction of the claim of providing the first precise result (the updated submission refers to "Gaussian posterior approximations" and "average Laplace approximation" around L149, which is not precise, at least calling it precise without defining the exact notion of precision can be confusing)
- A self-critical assessment of the assumptions, including the ones omitted, compared to prior works

**Additional Comments On Reviewer Discussion:**

As detailed above, there was a robust discussion between the authors and reviewers, primarily centered around arguing the novelty of the work compared to [1].  See above.

---

### Decision · Program_Chairs · 2025-01-22

Reject